# PPTC7 maintains mitochondrial protein content by suppressing receptor-mediated mitophagy

Natalie M. Niemi [1,2] ✉, Lia R. Serrano[3], Laura K. Muehlbauer[4], Catherine E. Balnis[3], Lianjie Wei[2], Andrew J. Smith [5], Keri-Lyn Kozul[6], Merima Forny[2], Olivia M. Connor[7], Edrees H. Rashan[8], Evgenia Shishkova [3,9], Kathryn L. Schueler[8], Mark P. Keller [8], Alan D. Attie [8], Jonathan R. Friedman [7], Julia K. Pagan[6,10,11], Joshua J. Coon [1,3,4,9] & David J. Pagliarini [1,2,5,8,12] ✉

PPTC7 is a resident mitochondrial phosphatase essential for maintaining proper mitochondrial content and function. Newborn mice lacking *Pptc7* exhibit aberrant mitochondrial protein phosphorylation, suffer from a range of metabolic defects, and fail to survive beyond one day after birth. Using an inducible knockout model, we reveal that loss of *Pptc7* in adult mice causes marked reduction in mitochondrial mass and metabolic capacity with elevated hepatic triglyceride accumulation. *Pptc7* knockout animals exhibit increased expression of the mitophagy receptors BNIP3 and NIX, and *Pptc7*[-/-] mouse embryonic fibroblasts (MEFs) display a major increase in mitophagy that is reversed upon deletion of these receptors. Our phosphoproteomics analyses reveal a common set of elevated phosphosites between perinatal tissues, adult liver, and MEFs, including multiple sites on BNIP3 and NIX, and our molecular studies demonstrate that PPTC7 can directly interact with and dephosphorylate these proteins. These data suggest that *Pptc7* deletion causes mitochondrial dysfunction via dysregulation of several metabolic pathways and that PPTC7 may directly regulate mitophagy receptor function or stability. Overall, our work reveals a significant role for PPTC7 in the mitophagic response and furthers the growing notion that management of mitochondrial protein phosphorylation is essential for ensuring proper organelle content and function.

Mitochondria engage in numerous cellular processes in eukaryotic cells, including ATP synthesis via oxidative phosphorylation, cofactor biosynthesis, and the commitment to cell death. Over time, mitochondria can become damaged, requiring quality control mechanisms to maintain a healthy mitochondrial population[1]. One such mechanism is mitophagy, which engages the autophagic machinery to clear damaged or superfluous organelles[2]. A growing body of work suggests that protein phosphorylation plays a key role in regulating mitophagy[3], and in various other mitochondrial processes such as protein import[4,5], core metabolism[6,7], heme biosynthesis[8], and programmed cell death[9]. Indeed, multiple candidate protein phosphatases reside at or within mitochondria[10–13] where they would be positioned to manage phosphorylation on mitochondrial proteins. Despite their prevalence, many of these phosphatases remain poorly studied.

Our recent work demonstrates that loss of one phosphatase, PPTC7, leads to the hyperphosphorylation of select mitochondrial proteins and significant loss of overall mitochondrial protein content[14]. *Pptc7* knockout (KO) animals present with hypoketotic hypoglycemia and fail to survive a single day after birth. *Pptc7* knockout tissues show widespread decreases in mitochondrial protein levels with little to no effects on matched mRNA levels, suggesting altered post-transcriptional control of mitochondrial content.

To further explore this phenomenon, we generated two independently derived *Pptc7* knockout models: a conditional model that allows inducible *Pptc7* knockout in adult mice, and isolated fibroblasts from our global *Pptc7* (i.e., perinatal lethal) knockout mouse model. We find that knockout of *Pptc7* decreases mitochondrial protein content in all tested systems. These molecular changes were accompanied by metabolic defects, including diminished oxygen consumption in cells exposed to multiple nutrients, as well as hepatic lipid accumulation in vivo. Interestingly, the mitophagy receptors BNIP3 and NIX were among the limited number of proteins that increased in expression across both mouse models and isolated fibroblasts. Our recent MITOMICS study revealed that *PPTC7* loss also drives elevation of these receptors in human cells, and this response was unique across over 200 knockout cell lines[15]. CRISPR-mediated knockout of BNIP3 and NIX (gene name *Bnip3l*) rescued the expression of most mitochondrial proteins in *Pptc7* KO cells, demonstrating that these receptors are necessary for the observed mitophagy. However, mitochondrial dysfunction persisted in *Pptc7* knockout cells after *Bnip3* and *Bnip3l* knockout, indicating that this dysfunction is connected at least partially to the underlying aberrant protein phosphorylation. Our phosphoproteomic analyses suggest a varied set of substrates for the phosphatase PPTC7, including the import protein TIMM50, multiple core metabolic proteins, and the BNIP3 and NIX mitophagy receptors themselves. Collectively, these data demonstrate that PPTC7 is critical for managing mitochondrial protein phosphorylation as well as regulating BNIP3- and NIX-associated mitophagy.

## Results

### PPTC7 maintains mitochondrial protein levels in adult mice

We demonstrated previously that knockout (KO) of the mitochondrial protein phosphatase *Pptc7* causes decreased mitochondrial content, metabolic dysfunction, and fully penetrant lethality within one day of birth[14]. To overcome the limitations associated with these severe phenotypes, we generated a floxed *Pptc7* mouse with the potential for conditional knockout (Fig. 1a). We bred floxed *Pptc7* mice to the well-established UBC-Cre-ER[T2] mouse[16] to allow widespread, inducible recombination in response to tamoxifen (Fig. 1b). We validated tamoxifen-induced genomic excision of *Pptc7* exon 3 across seven tissues (Supplementary Fig. 1A), which we confirmed at the protein level in liver (Supplementary Fig. 1B). Furthermore, age-matched female Cre-positive floxed animals did not show significant levels of recombination in the absence of tamoxifen (Supplementary Fig. 1A), indicating that "leaky" Cre-induced knockout of *Pptc7* is not a significant concern through the timeframe of our study.

Upon confirming tamoxifen-inducible KO of *Pptc7* in adult mice, we isolated liver tissue from control (i.e., UBC-Cre[+/-];*Pptc7*[+/+]) and knockout (i.e., UBC-Cre[+/-];*Pptc7*[flox/flox]) mice of both sexes two weeks post-tamoxifen administration and analyzed these tissues via 16-plex tandem mass tag (TMT) quantitative proteomics (Fig. 1c). In this experiment, we identified 6749 proteins, 1367 of which were altered in *Pptc7*[-/-] liver relative to control tissue. Approximately equal numbers of proteins significantly increased (646) or decreased (721) in abundance ($p < 0.05$, Student's *t* test). Notably, when stratified for mitochondrial localization, only 14 proteins in MitoCarta 3.0[13] exhibited a $\log_2$ fold change of >0.2 in *Pptc7* KO liver, while 512 mitochondrial proteins exceeded a $\log_2$ fold change of −0.2

(Supplementary Fig. 1C, Source data 1). These results demonstrate that acute loss of *Pptc7* leads to decreased mitochondrial protein content in adult mouse liver comparable to what we observed in the perinatal global *Pptc7* KO model[14] (Fig. 1d, e). Importantly, PPTC7 was the most significantly downregulated mitochondrial protein in our proteomics analysis (Fig. 1d), providing confirmation of *Pptc7* knockout in UBC-Cre[+/-];*Pptc7*[flox/flox] liver tissue. We additionally confirmed decreased mitochondrial content through measurements of citrate synthase (CS) activity (Supplementary Fig 1D), CS protein expression (Supplementary Fig 1E), and mitochondrial DNA (mtDNA) levels (Supplementary Fig 1F). Despite the trend of decreased mitochondrial markers, a few proteins, including BNIP3 and its paralog, NIX, were significantly elevated in *Pptc7* KO liver (Fig. 1d). An analysis of the proteomic changes between our two KO models revealed significant correlation (Fig. 1e), suggesting PPTC7 plays a similar role in liver tissue across developmental states. Collectively, these data demonstrate that acute loss of *Pptc7* in adult mouse liver globally decreases mitochondrial content, suggesting that PPTC7 expression is required across at least two physiological states to maintain mitochondrial protein homeostasis.

To investigate the physiological response to acute *Pptc7* loss, we performed gene function analysis on our proteomics dataset. This analysis revealed a significant enrichment in proteins involved in fatty acid oxidation (FAO, Fig. 1f), suggesting *Pptc7* KO animals may have disrupted lipid metabolism. As global *Pptc7* KO animals exhibit signs of diminished FAO such as hypoketosis[14], we tested whether adult mice with acute loss of *Pptc7* showed similar defects. We found that *Pptc7* KO male mice did not have significantly altered ketone levels in response to an overnight fast (Fig. 1g) but that both sexes of *Pptc7*[-/-] mice had significantly elevated liver triacylglycerols (TAGs) (Fig. 1h), consistent with hepatic steatosis. Interestingly, male *Pptc7* KO mice also showed significantly increased expression of multiple perilipin proteins, which associate with lipid droplets and are elevated during steatosis[17,18] (Fig. 1i). We also examined hepatic and circulating cholesterol levels, as well as serum TAGs, which did not significantly change in *Pptc7* KO animals relative to control littermates (Supplementary Fig. 1G–I). Collectively, these data suggest that acute loss of *Pptc7* perturbs hepatic lipid homeostasis but in a manner distinct from that observed in the perinatal KO.

### Loss of Pptc7 broadly decreases mitochondrial metabolism in isolated cells

To complement our studies in mice, we established a cell-based system to interrogate mechanisms driving the metabolic and mitochondrial abnormalities in *Pptc7* KO models. We bred *Pptc7*[+/-] mice from our global KO model to generate mouse embryonic fibroblasts (MEFs) from three wild-type and three knockout E14.5 embryos, which were validated at the gene (Fig. 2a) and protein (Fig. 2b) levels. A proteomic analysis of these cells revealed significant mitochondrial protein loss (Fig. 2c) consistent with our observations from the global *Pptc7* KO heart and liver tissues and inducible *Pptc7* KO liver tissue (although the relative reduction of mitochondrial proteins in vivo is larger than that seen in the MEFs) (Fig. 2d). These data demonstrate that loss of *Pptc7* decreases mitochondrial protein content cell autonomously and in a third distinct cell or tissue type.

We performed correlation analysis of all overlapping proteins identified in MEFs, perinatal heart tissue, and perinatal liver tissue, and found significant positive correlation between *Pptc7* KO models (Supplementary Fig. 2A, B). These data also reveal that proteins involved in metabolic pathways, such as oxidative phosphorylation, the TCA cycle, and fatty acid oxidation, were more strongly diminished in *Pptc7* KO cells and tissues (Supplementary Fig. 2C–E) than other non-metabolic pathways (e.g., the mitochondrial ribosome, Supplementary Fig. 2F–H). To test whether *Pptc7* KO cells have compromised metabolism, we performed Seahorse assays and found that primary

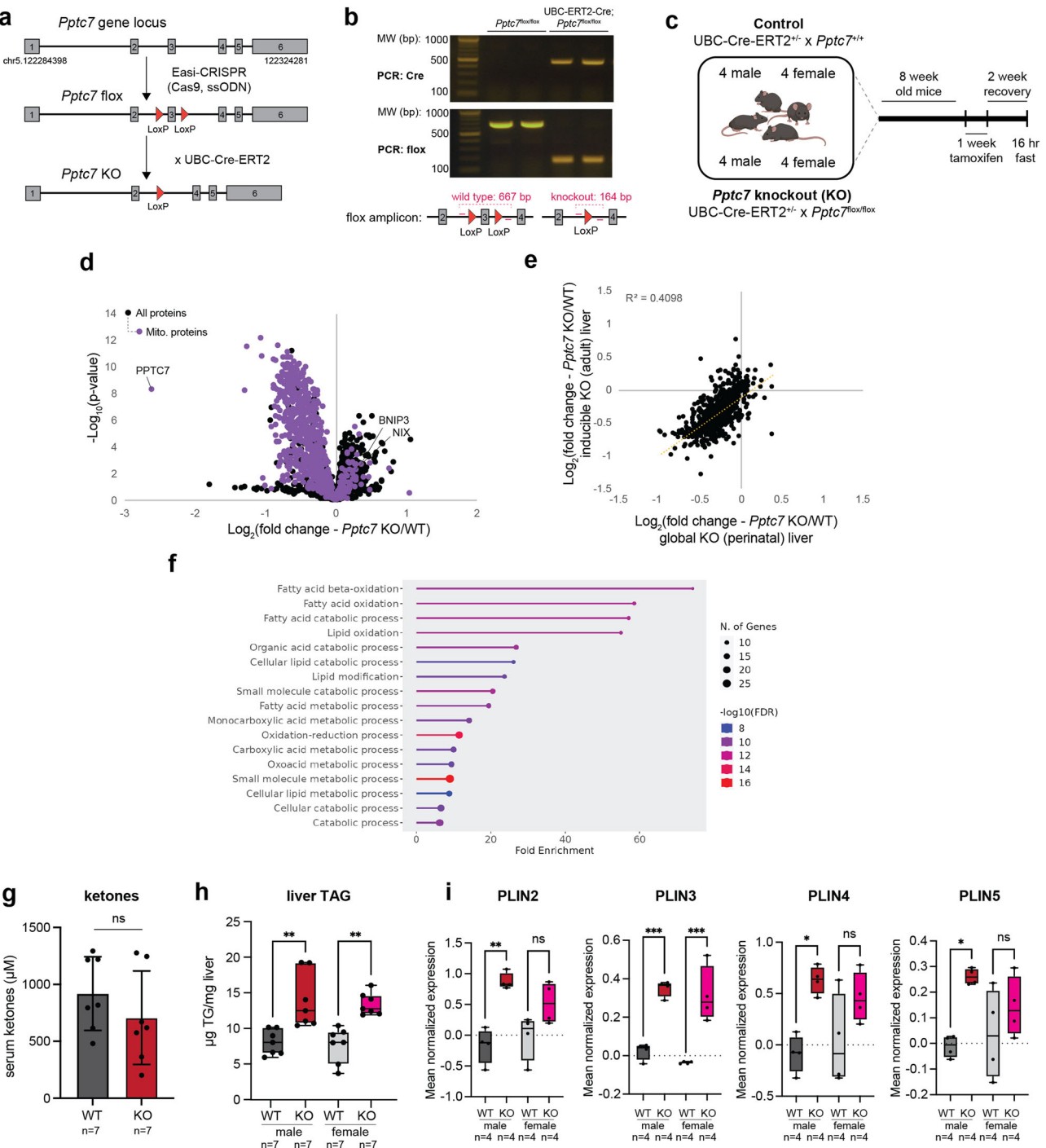

*Pptc7* KO fibroblasts have mildly decreased oxygen consumption rates (OCR) in basal conditions but have substantially lower spare respiratory capacity than wild-type fibroblasts (Fig. 2e). We also tested OCR in permeabilized wild-type and *Pptc7* KO fibroblasts provided with pyruvate/malate or succinate/rotenone and found that KO cells given either substrate treatment showed decreased basal OCR and spare respiratory capacity (Fig. 2f, g). As our data suggested that acute loss of *Pptc7* compromises fatty acid oxidation in mouse liver (Fig. 1), we also performed Seahorse analysis of MEFs exposed to palmitate-BSA in the presence or absence of the CPT1 inhibitor etomoxir. These data show that palmitate-BSA induces a significant increase in OCR in wild-type but not *Pptc7* KO MEFs (Fig. 2h), consistent with deficient fatty acid oxidation in the latter. As an orthogonal approach, we performed a BioLog assay to test the capacity of *Pptc7* KO cells to catabolize various

nutrients. These experiments showed that *Pptc7* KO cells have a compromised ability to metabolize substrates that feed into the TCA cycle (e.g., pyruvate, α-ketoglutarate, succinate, fumarate, or malate, Fig. 2i), glutamine catabolism (e.g., glutamate, glutamine, and Glutamax, Fig. 2j), and fatty acid catabolism (e.g., octanoyl-carnitine, Fig. 2i). Collectively, these data demonstrate that PPTC7 expression is required to maintain mitochondrial protein levels in isolated cells and that loss of this phosphatase disrupts multiple metabolic pathways.

## Loss of mitochondrial content is mediated by the mitophagy receptors Bnip3 and Nix

The consistent loss of mitochondrial protein levels across each *Pptc7* KO tissue and cell type (Fig. 2d) suggests a common molecular driver for this phenotype. The mitophagy receptor BNIP3 is significantly

**Fig. 1 | Acute knockout of Pptc7 compromises hepatic mitochondrial content in adult mice. a** Schematic for the generation of a conditional *Pptc7* model to allow global, inducible knockout in adult mice. **b** Genotyping verification of Cre-mediated excision of *Pptc7* exon 3 after tamoxifen treatment. This experiment is representative of at least three independent experiments. **c** Schematic for the 16-plex proteomic analysis of liver tissue from control (UBC-ER^T2-Cre;*Pptc7*^+/+) or experimental (UBC-ER^T2-Cre;*Pptc7*^flox/flox) animals (*n* = 4 for each sex and genotype for 16 total). **d** Proteomic analysis of non-mitochondrial (black dots) and mitochondrial (purple dots) proteins across 16 liver samples from mice aged 11 weeks, 2 weeks post-tamoxifen treatment. Data were analyzed via a two-sided Student's *t* test with log-transformed *p*-values reported on the y-axis. **e** Linear regression of the fold changes in mitochondrial proteins (*n* = 599) identified in both adult inducible KO liver (y-axis, *n* = 8 control and *n* = 8 knockout tissues) and perinatal KO liver (x-axis, *n* = 5 wild-type and *n* = 5 knockout tissues). Regression coefficient reported as R². **f** GO term analysis of non-mitochondrial proteins that are altered in *Pptc7* knockout inducible liver (*n* = 8 knockout tissues) relative to wild-type (*n* = 8 control tissues) animals. **g** Serum ketones quantified in male wild-type (WT, *n* = 7 animals, gray bar) and *Pptc7* knockout (KO, *n* = 7 animals, red bar) mice aged 11 weeks, 2 weeks post-tamoxifen treatment, fasted overnight. A two-sided Student's *t* test was performed, ns = not significant. The bar graph center denotes the mean of the

dataset; error bars represent standard deviation. **h** Liver triacylglycerol content (TAG) in male and female wild-type (WT (*n* = 7 male WT, dark gray bar and *n* = 7 female WT, light gray bar) and *Pptc7* KO (*n* = 7 KO male, red bar and *n* = 7 KO female, pink bar) mice aged 11 weeks, 2 weeks post-tamoxifen treatment, fasted overnight. Ordinary one-way ANOVA performed. **\*\****p* < 0.01. *p*-value for male WT v. KO comparison = 0.0012; *p*-value for female WT v. KO comparison = 0.0013. The box plot extends from the 25th to 75th percentile; whiskers stretch from minimum to maximum datapoints. The line in the middle of the box plot represents the median. **i** Protein expression, quantified via LC-MS, of perilipin proteins (PLINs) in male and female wild-type (WT) and *Pptc7* KO mice aged 11 weeks, 2 weeks post-tamoxifen treatment. Ordinary one-way ANOVA performed. \*\*\**p* < 0.001, \*\**p* < 0.01, \**p* < 0.05, ns = not significant. For PLIN2 measurements, *p*-value for male WT v. KO = 0.0013; *p*-value for female WT v. KO comparison = 0.0887; for PLIN3 measurements, *p*-value for male WT v. KO = 0.0003; for female WT v. KO comparison = 0.0002; for PLIN4 measurements, *p*-value for male WT v. KO = 0.01323; *p*-value for female WT v. KO comparison = 0.1889; for PLIN5 *p*-value for male WT v. KO = 0.0161; *p*-value for female WT v. KO comparison = 0.5668. The box plot extends from the 25th to 75th percentile; whiskers stretch from minimum to maximum datapoints. The line in the middle of the box plot represents the median. Source data are provided as a Source data file. Figure 1c was created with BioRender.

upregulated across all cell types and tissues assayed, with some systems, including the MEFs, showing additional upregulation of NIX (gene name *Bnip3l*), a BNIP3 paralog[14] (Figs. 1c and 2d). We hypothesized that BNIP3 and NIX could promote excessive mitophagy, contributing to the decrease in mitochondrial protein levels seen in *Pptc7* KO systems. To test this, we used CRISPR/Cas9 to disrupt *Bnip3* and *Bnip3l* in the *Pptc7* knockout background, which we validated by protein expression (Fig. 3a). We transduced these cells with the mt-Keima mitophagy reporter, a pH-dependent fluorescent protein that exhibits distinct emission spectra in mitochondria at physiological pH and in the acidic lysosomal compartment after mitophagy[19]. *Pptc7* KO cells expressing mt-Keima showed elevated numbers of acidic mitochondria via both microscopy (Fig. 3b, c) and flow cytometry (Fig. 3d and Supplementary Fig. 3A), indicating excess mitophagic flux. Importantly, this increase was abolished following CRISPR/Cas9-based disruption of *Bnip3* and *Bnip3l*, as seen in both microscopy (Fig. 3b, c) and FACS-based approaches (Fig. 3d and Supplementary Fig. 3A). Conversely, mitophagy was elevated when BNIP3 or NIX were over-expressed in wild-type MEFs (Supplementary Fig. 3B, C). Importantly, analysis of proteomics data in both *Pptc7* KO liver tissue and MEFs found only BNIP3 and NIX to be significantly upregulated among identified proteins within the receptor-mediated or ubiquitin-mediated mitophagy pathways (Supplementary Fig. 3D, E). Most proteins within the autophagic pathway do not significantly change in expression in response to *Pptc7* KO (Supplementary Fig. 3D, E). However, activated LC3-II accumulated to a greater extent in *Pptc7* KO MEFs treated with bafilomycin A relative to wild-type cells (Supplementary Fig. 3F), suggesting higher autophagic flux. Collectively, these data suggest that BNIP3 and NIX drive the elevated mitophagy flux seen in *Pptc7* knockout cells. Consistently, proteomic analysis of each of the *Pptc7/Bnip3/Bnip3l* knockout clones (herein referred to as TKO clones), as well as double *Bnip3/Pptc7* and *Bnip3l/Pptc7* KO clones revealed substantial rescue of mitochondrial proteins relative to *Pptc7* KO cells (Fig. 3d).

We next tested whether the knockout of *Bnip3* and *Bnip3l* rescued the metabolic defects seen in *Pptc7* KO cells. Basal oxygen consumption was partially rescued in both TKO clones, but spare respiratory capacity was not rescued in either TKO clone relative to *Pptc7* KO cells (Fig. 3e). Furthermore, the TKO cells exhibited diminished OCR and ECAR relative to wild-type MEFs (Fig. 3f), similar to the *Pptc7* KO cells. These data demonstrate that while the dual knockout of BNIP3 and NIX reversed mitochondrial protein loss in the *Pptc7* KO cells, it did not rectify the underlying metabolic defects.

Due to the persistent metabolic defects seen in *Pptc7/Bnip3/Bnip3l* TKO cells, we considered the possibility that stabilization of BNIP3 and NIX may be an indirect, common downstream consequence of general mitochondrial dysfunction. To assess this, we analyzed the data from our recent MITOMICS study, in which we performed multiomic analyses on over 200 HAP1 cell lines carrying monogenic deletions of genes encoding mitochondria-localized proteins. Remarkably, *Pptc7* was the only gene whose KO resulted in significant increase of both BNIP3 and NIX expression in these cells (Fig. 3g, h). These data suggest that upregulation of BNIP3 and NIX is not a general response to mitochondrial stress (at least in HAP1 cells), but rather that these mitophagy receptors are selectively elevated in response to the loss of *Pptc7*.

## Phosphoproteomic analyses reveal BNIP3 and NIX as candidate PPTC7 substrates

Our analyses above suggest that the metabolic aberrations caused by PPTC7 loss are rooted in dysregulated protein phosphorylation that then leads to BNIP3/NIX-mediated mitophagy. Our new model systems offer an opportunity to identify common PPTC7 substrates across cells and tissues. To identify these putative substrates, we performed phosphoproteomic analyses on liver tissue from both sexes of mice from our inducible *Pptc7* KO model (*n* = 16 mice; 4 per sex and genotype, Fig. 4a) and matched wild-type and *Pptc7* KO MEFs (*n* = 3 independent cell lines for wildtype and KO, respectively, Fig. 4b). As expected from the disruption of a protein phosphatase, the majority of significantly altered mitochondrial phosphoisoforms were elevated in *Pptc7* KO samples across both systems (Fig. 4a, b), whereas similar trends were not seen in the non-mitochondrial phosphoproteome (Supplementary Fig. 4A, B). The results, especially when combined with our earlier data[14], suggest a complex regulatory picture. Collectively, hundreds of mitochondrial phosphoproteins are elevated across all systems, including marked changes on multiple metabolic proteins. However, only four phosphoisoforms are significantly elevated across all systems (Fig. 4c). Of note, there is substantially higher overlap of quantified phosphoisoforms in perinatal and adult model liver tissue from *Pptc7* KO mice than between other samples (Fig. 4d), suggesting tissue specificity for PPTC7 substrates. These data are consistent with a recent study that found significant functional specialization of mitochondrial phosphoproteomes across tissues in mice[20]. Validation of these potential substrates and their contribution to the *Pptc7* KO metabolic phenotype across tissues will require further mechanistic investigations.

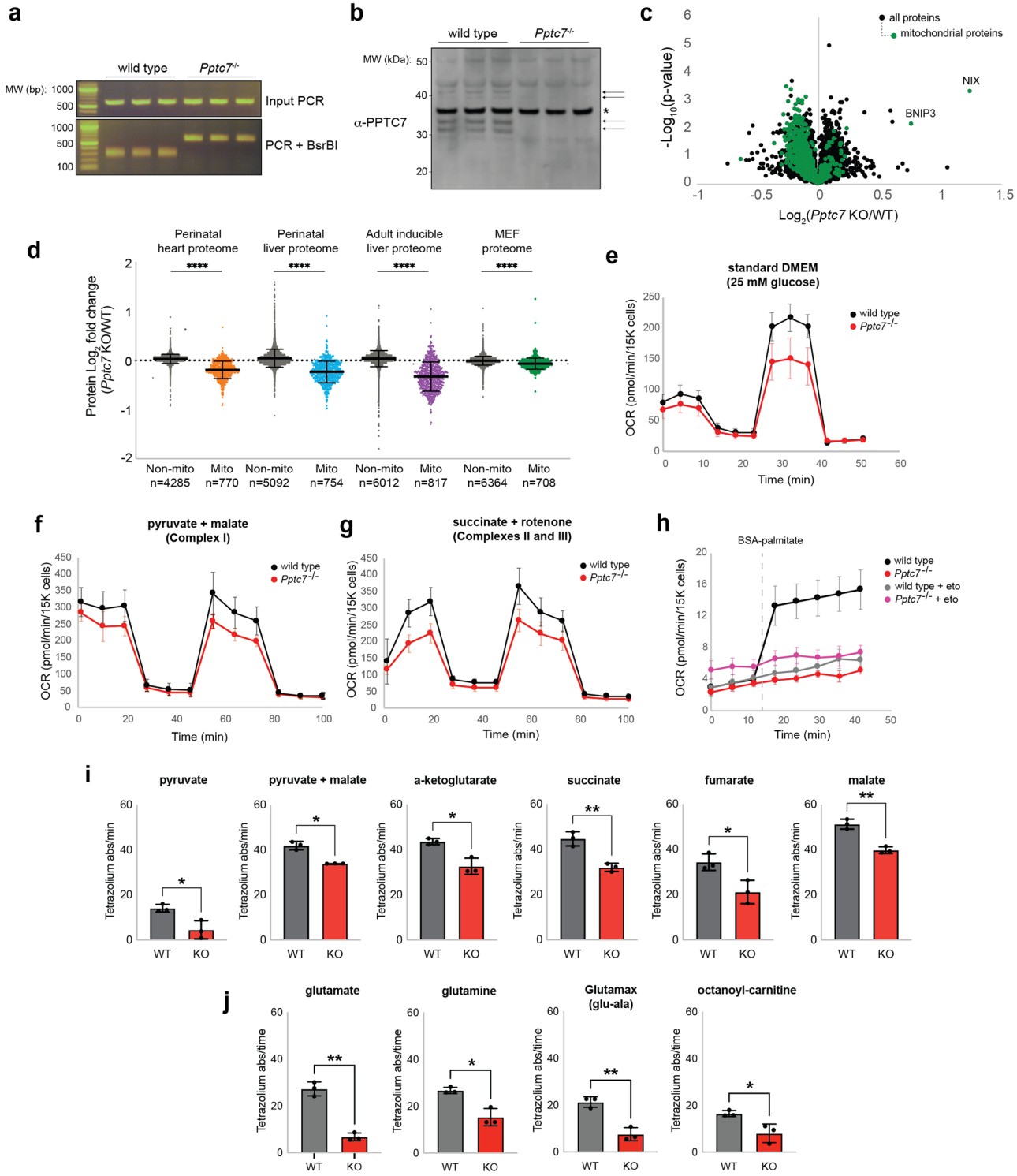

Notably, we identified multiple phosphorylation sites on the BNIP3 and NIX receptors themselves. These phosphorylation sites do not exhibit significant increases in the *Pptc7* KO cells when the data are subject to the standard normalization to total protein levels. However, this normalization could mask meaningful hits in instances whereby the phosphosite in question promotes protein stability and is found at high stoichiometry. In this case, loss of a phosphatase could result in the accumulation of stabilized, phosphorylated substrates that is not evident by a change in apparent phosphorylation occupancy. Given this, and the substantial upregulation of BNIP3 and NIX in our systems, we examined our phosphoproteomics data without total protein

normalization and found multiple elevated phosphopeptides for both BNIP3 and NIX across all *Pptc7* KO systems (Fig. 4e). These phosphorylation events span eight unique residues in each of the BNIP3 and NIX proteins (Fig. 4f), suggesting complex phosphorylation-based regulation. Two of these observed sites, S60 and T66, were identified across each of our experimental systems (Fig. 4g) and were recently linked to BNIP3 stability[21], adding further credence to the possibility that these phosphoisoforms may largely be representative of the total cellular pool of these proteins. Many aspects of this regulation remain to be elucidated, including the activity of kinase(s) responsible for BNIP3 and NIX phosphorylation, the effect of phosphorylation on

**Fig. 2 | *Pptc7* knockout causes cell-autonomous decreases in mitochondrial protein content leading to broad metabolic defects. a** Genotyping of wild-type (WT) and *Pptc7* KO mouse embryonic fibroblasts (MEFs). This experiment is representative of at least three independent experiments**. b** PPTC7 endogenous protein expression in WT and KO MEFs. Notably, PPTC7 runs as a set of two doublets, marked by arrows. A non-specific band (*) serves as a loading control. This experiment is representative of at least three independent experiments.
**c** Proteomic analysis of non-mitochondrial (black dots) and mitochondrial (green dots) proteins from triplicate WT and *Pptc7* KO MEFs. Data were analyzed via a two-sided Student's *t* test with log-transformed *p*-values reported on the y-axis. The mitophagy receptors BNIP3 and NIX are highlighted. **d** Fold changes of mitochondrial proteins across all *Pptc7* KO systems including perinatal heart (orange) and liver (blue), inducible adult liver (purple), and MEFs (green). In each system, loss of *Pptc7* causes significant decreases in the mitochondrial proteome relative to non-mitochondrial proteins (shown in gray for all systems; total n for each dataset listed below graph). ****$p < 0.0001$, ordinary one-way ANOVA; multiple comparisons across each paired WT and KO dataset. **e** Seahorse analysis of a mitochondrial

stress test performed on primary wild-type (black dots) and *Pptc7* KO (red dots) MEFs grown in standard DMEM supplemented with 25 mM glucose. Data are represented as the mean of *n* = 9 replicate wells per condition; error bars represent standard deviation. **f, g** Seahorse analysis of a mitochondrial stress test performed on primary permeabilized wild-type (black dots) and *Pptc7* KO (red dots) MEFs given pyruvate and malate (**f**) or succinate and rotenone (**g**). The data are represented as the mean of *n* = 8 replicate wells per condition. Error bars represent standard deviation. **h** Seahorse analysis of wild-type (black or gray dots) or *Pptc7* KO (red and pink dots) MEFs given BSA-palmitate in the presence or absence of of 4 μM etomoxir. Data are represented as the mean of *n* = 8 replicate wells per condition. Error bars represent standard error of the mean. **i, j** BioLog analysis of permeabilized wild-type (gray bars) and *Pptc7* KO (red bars) MEFs given various TCA cycle substrates (**i**) or amino acids and fatty acids (**j**). For BioLog analysis, each datapoint represents an independent well on the BioLog plate (*n* = 3), with the mean shown. Error bars represent standard deviation. *$p < 0.05$, **$p < 0.01$; BioLog data was analyzed using a two-sided Student's *t* test. Source data are provided as a Source data file.

protein half-life, and the absolute phosphorylation stoichiometry. Nonetheless, the consistent upregulation of these phosphopeptides in *Pptc7* KO systems suggests that PPTC7 influences this phosphorylation status, possibly through direct interaction.

### PPTC7 can directly interact with and dephosphorylate BNIP3 and NIX

We previously performed a large-scale affinity enrichment-mass spectrometry analysis using mitochondrial baits in two human cell lines[22]. In reanalyzing these data, we discovered that BNIP3 and NIX were the strongest interaction partners identified for PPTC7 in both 293T and HepG2 cells (Fig. 5a). Notably, of the 77 baits tested, PPTC7 is the only bait to capture endogenous BNIP3 or NIX beyond significance thresholds, suggesting that this interaction is both specific and reproducible across experiments and cell types. To confirm these results, we performed immunoprecipitations of overexpressed PPTC7-FLAG in U2OS cells. Endogenous BNIP3 and NIX are kept at low basal expression levels, resulting in difficulty capturing PPTC7-FLAG via immunoprecipitation in basal conditions (Fig. 5b). However, upon upregulation of BNIP3 and NIX through treatment with deferiprone (DFP), an iron chelator well known to induce BNIP3 and NIX[23], PPTC7 robustly immunoprecipitated both mitophagy receptors (Fig. 5b), further supporting a strong interaction between these proteins.

While these immunoprecipitations demonstrate that PPTC7 can interact with BNIP3 and NIX, it remained unclear whether these interactions were direct. To address this, we performed yeast two-hybrid analyses to analyze interactions between human PPTC7, BNIP3, and NIX. These data demonstrated strong interactions between PPTC7 and each mitophagy receptor in a reciprocal fashion (Fig. 5c), suggesting direct interaction. To confirm these interactions, we generated recombinant PPTC7 to query whether this phosphatase could directly dephosphorylate BNIP3 and NIX. We reasoned that such an experiment would test whether PPTC7 can directly and functionally interact with BNIP3 and NIX, as well as test whether PPTC7 has selectivity toward the phosphosites identified in our analyses. We isolated crude mitochondria from *Pptc7* KO MEFs, which have elevated phosphorylation on multiple residues on BNIP3 and NIX (Fig. 4). We incubated these isolated mitochondria with either wild-type or catalytically inactive recombinant PPTC7 and subsequently performed phosphoproteomic analyses. These data demonstrate that wild-type, but not catalytically inactive, PPTC7 dephosphorylated BNIP3 (Fig. 5d) and NIX (Fig. 5e) at most identified phosphosites. Collectively, these data demonstrate that PPTC7 can directly interact with BNIP3 and NIX to promote their dephosphorylation. Whether these interactions are maintained in vivo and under the conditions in which dephosphorylation may occur within intact cells should be an area of active investigation in the future.

## Discussion

Our data demonstrate that *Pptc7* knockout leads to BNIP3 (and, in some systems, NIX) upregulation, excessive mitophagy, and decreased steady state levels of mitochondrial proteins. BNIP3 and NIX are phosphorylated in *Pptc7* knockout systems, consistent with a model in which phosphorylation promotes BNIP3- and NIX-mediated mitophagic signaling. Notably, in *S. cerevisiae*, the single mitophagy receptor Atg32 is phosphorylated by casein kinase 2 (CK2) to promote its interaction with Atg11 and subsequently induce mitophagy[24,25]. Similar mechanisms activate mammalian mitophagy receptors, including FUNDC1, which is phosphorylated by CK2 and dephosphorylated through the phosphatase PGAM5[26], and the Atg32 homolog BCL2L13, whose phosphorylation allows recruitment of the mitophagy machinery[27]. Our data suggest that similar activation mechanisms may exist for BNIP3 and NIX, albeit ones that are more complex and that are potentially mediated through diverse signaling inputs, as evidenced by the numerous phosphorylation events identified on each receptor. Importantly, phosphorylation of BNIP3 and NIX has been previously linked to the function and stability of these proteins. Similar to FUNDC1, phosphorylation near the N-terminal LC3 interaction region (LIR) of BNIP3 and NIX stabilizes their interaction with LC3 to promote mitophagy[28–30]. Phosphorylation has also been identified within the C-terminal regions of BNIP3 and NIX, influencing the ability of BNIP3 to promote cell death[31] and the subcellular localization of NIX[32]. Finally, a recent study found that phosphorylation of BNIP3 at S60 and T66−two residues identified in our analysis−stabilizes this receptor by blocking its ubiquitination-mediated turnover[21]. Three independent studies recently identified the mitochondrial E3 ligase FBXL4 as responsible for the basal turnover of BNIP3 and NIX[33–35]. Knockout of *Fbxl4* increases BNIP3 and NIX protein levels and promotes excessive mitophagy, leading to diminished mitochondrial protein levels and perinatal lethality in mice[36]−phenotypes that strikingly mirror those observed in our *Pptc7* knockout models[14].

Accumulating evidence suggests that cells devote significant resources to the proper regulation of BNIP3 and NIX activity. TMEM11, a transmembrane protein linked to the MICOS complex, was recently found to physically interact with BNIP3 and NIX and mediate their spatial localization[37]. Importantly, knockout of TMEM11 triggers hyperactive mitophagy, likely due to an increased number of LC3-positive mitophagosome formation sites at the outer mitochondrial membrane[37]. These data suggest that the physical localization of BNIP3 and NIX is likely important for the proper regulation of mitophagy. Beyond this, BNIP3 protein levels seem to be actively limited in basal cell culture conditions, as BNIP3 is one of the shortest-lived mitochondrial proteins with a half-life of 103 min[38]. BNIP3 accumulates during proteasomal inhibition[21,28], suggesting efficient ubiquitin-mediated turnover. Consistently, *Fbxl4* knockout systems show

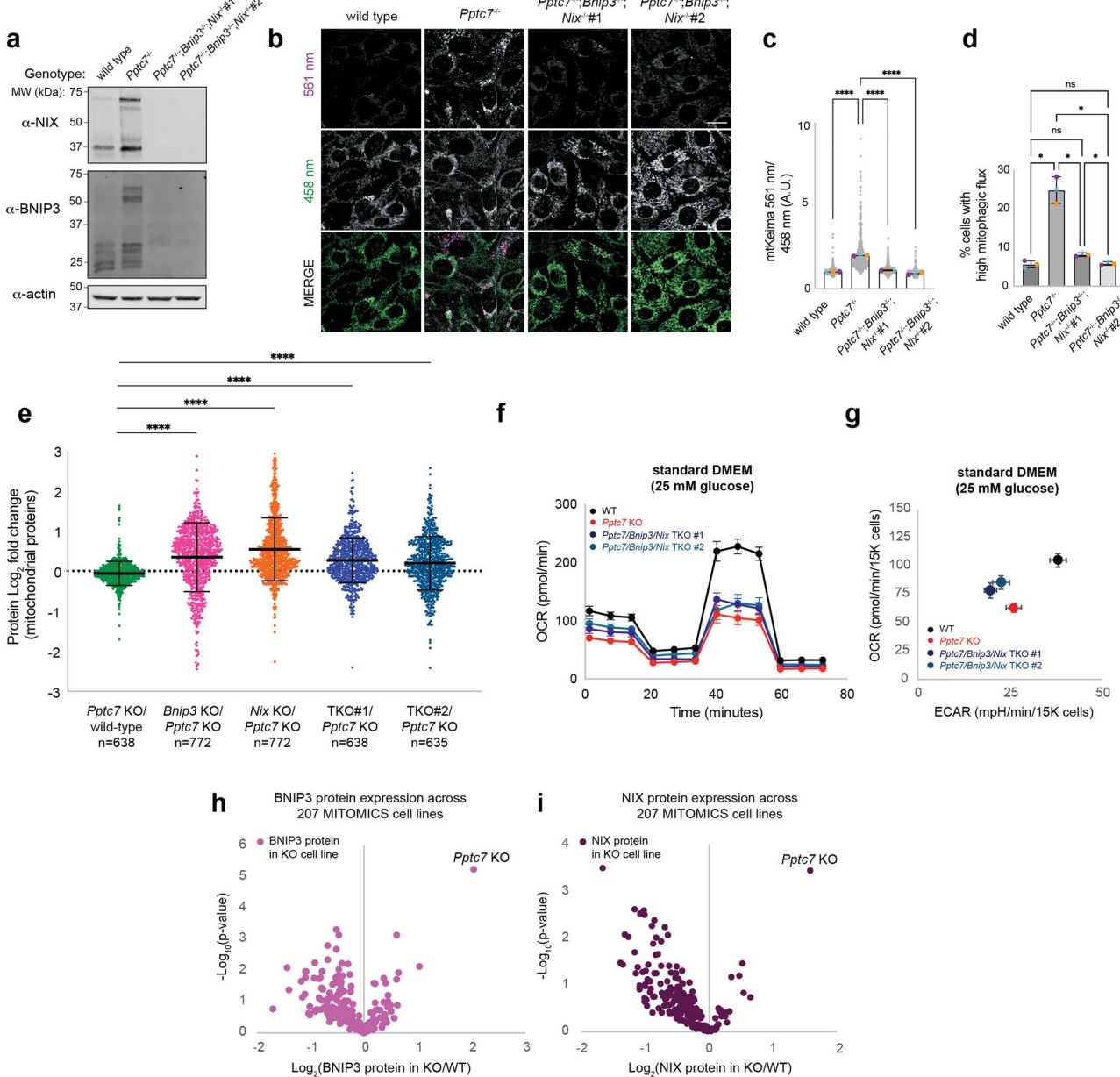

**Fig. 3 | *Pptc7* knockout causes excessive BNIP3- and NIX-mediated mitophagy.**
**a** Western blot of BNIP3 and NIX expression in wild-type, *Pptc7* KO, and two *Pptc7/Bnip3/Bnip3l* triple knockout (TKO) MEF cell lines. Actin is shown as a load control. This experiment is representative of at least three independent experiments.
**b** Representative images of wild-type, *Pptc7* KO, and each TKO cell line expressing the mitophagy reporter mt-Keima. Mitochondria imaged at 458 nm are at physiological pH but those imaged at 561 nm reflect acidic mitochondria undergoing mitophagy. **c** Quantification of mt-Keima imaging. Each gray dot represents the ratio of mitochondrial fluorescence from a single cell; $n = 123$ for wild type, $n = 152$ for *Pptc7* KO, $n = 123$ for TKO#1, $n = 143$ for TKO#2. Blue, orange, and purple dots represent averages from three independent biological experiments. The bar graphs intersect the mean of these three experiments; error bars represent standard deviation. ****$p < 0.0001$. mt-Keima microscopy data was analyzed by ordinary one-way ANOVA. **d** Quantification of mt-Keima as analyzed by FACS. Blue, orange, and purple dots represent averages from three independent biological replicates. The bar graphs intersect the mean of replicates; error bars represent standard deviation. mt-Keima microscopy data was analyzed by a Brown-Forsythe and Welch ANOVA. *$p < 0.05$, ns = not significant. For WT v. KO, $p = 0.0341$, for KO v. TKO #1,

$p = 0.0428$, for KO v. TKO #2, $p = 0.0332$. **e** Proteomic analysis of mitochondrial proteins in *Pptc7* knockout relative to wildtype (green), *Bnip3* knockout relative to *Pptc7* knockout (pink), *Nix* knockout relative to *Pptc7* knockout (orange), and two independent *Pptc7/Bnip3/Bnip3l* TKO lines normalized to *Pptc7* KO (dark blue and light blue). Each dot represents a quantified mitochondrial protein; n of each experiment is reported below the x-axis. Proteomic data was analyzed by ordinary one-way ANOVA. ****$p < 0.0001$. Mean and standard deviation shown. **f** Seahorse analysis of a mitochondrial stress test in immortalized wild type (WT, black), *Pptc7* KO (red), TKO #1 (dark blue), TKO #2 (teal) were assayed in Seahorse DMEM supplemented with 25 mM glucose. Data are represented as the mean of $n = 22$ replicate wells per condition; error bars represent standard deviation. **g** Seahorse analysis of basal oxygen consumption rates (OCR) and extracellular acidification (ECAR). Mean of $n = 8$ independent wells shown; error bars represent standard deviation. **h**, **i** BNIP3 (**h**) and NIX (**i**) protein expression across 207 cell lines harboring monogenic mutations in genes encoding mitochondria-localized proteins. Data were analyzed via a two-sided Student's *t* test with log-transformed *p*-values reported on the y-axis. Only *Pptc7* KO increases both BNIP3 and NIX significantly across this dataset. Source data are provided as a Source data file.

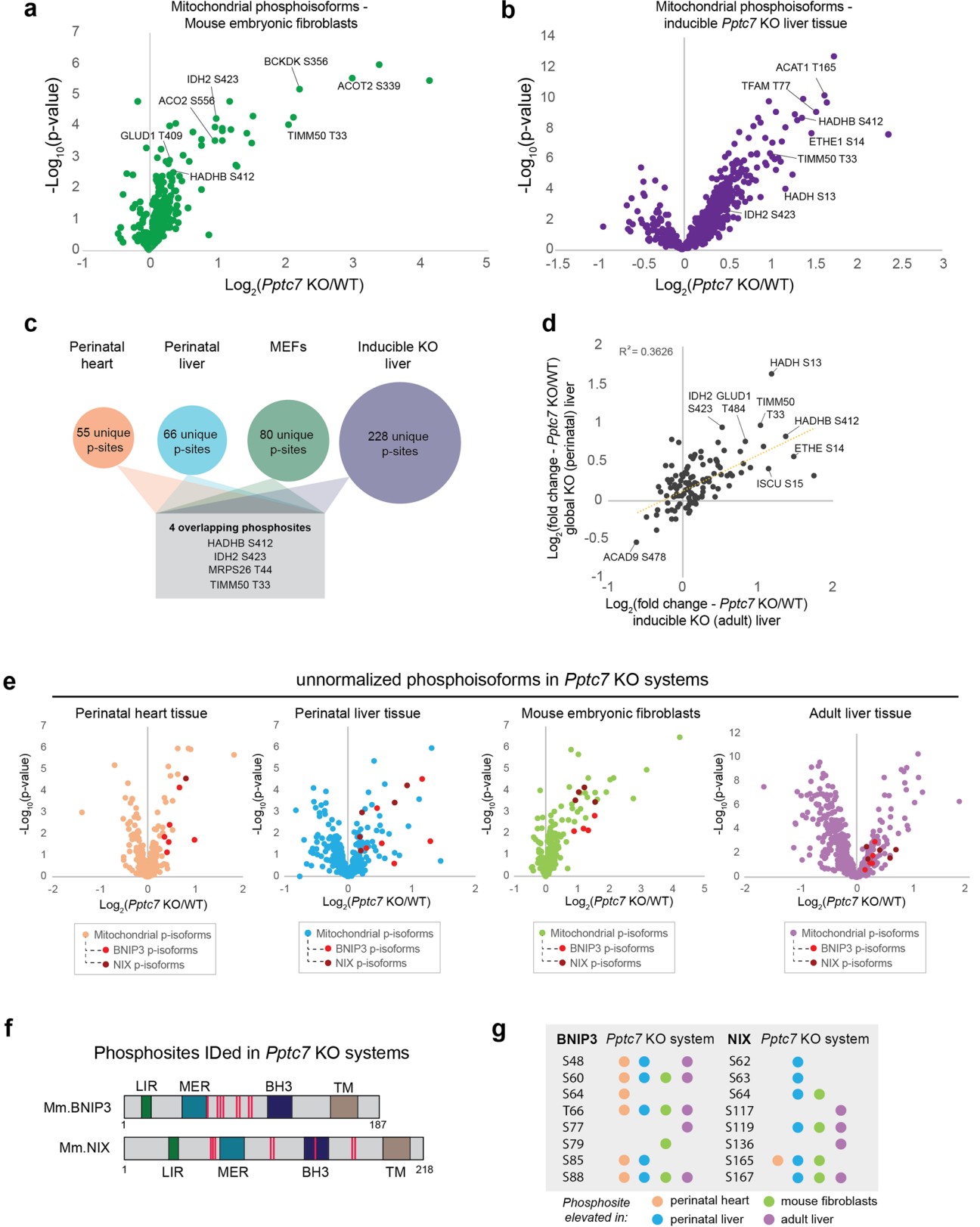

**e** unnormalized phosphoisoforms in *Pptc7* KO systems

**f** Phosphosites IDed in *Pptc7* KO systems

hyperactive basal mitophagy, further suggesting that efficient ubiquitination of BNIP3 and NIX is critical for maintaining mitochondrial content and proper organellar function[33–35]. Our data, as well as data generated by He et al., suggest that phosphorylation of BNIP3 (and likely NIX) may promote protein stability by blocking ubiquitination-mediated turnover. However, He et al. identified PP1/2A as the

phosphatase responsible for dephosphorylating BNIP3[21]. This leads to an important question: is PPTC7 directly responsible for the dephosphorylation of BNIP3 and NIX, or does it mediate these effects indirectly (for instance, through PP1/2A activity)?

Certain observations support each of these models. Our data suggest that PPTC7 interacts with BNIP3 and NIX as demonstrated by

**Fig. 4 | Phosphoproteomic analysis of *Pptc7* KO systems reveals candidate substrates, including BNIP3 and NIX. a, b** Protein-normalized mitochondrial phosphoisoforms in *Pptc7* KO MEFs (green, **a**) and inducible adult liver tissue from mice aged 11 weeks, 2 weeks post-tamoxifen treatment (purple, **b**). Data were analyzed via a two-sided Student's *t* test with log-transformed *p*-values reported on the y-axis. Select phosphorylation sites highlighted. **c** Analysis of phosphoproteomes across systems reveals many unique (i.e., only identified in a single experimental system) phosphosites (p-sites), with four significantly upregulated phosphosites identified across all four experimental systems (shown in gray box). **d** Linear regression analysis of overlapping phosphosites identified in perinatal liver tissue (y-axis) and inducible adult liver (x-axis). Select phosphorylation sites highlighted **e** Non-protein normalized mitochondrial phosphoisoforms were analyzed. Data were analyzed via a two-sided Student's *t* test with log-transformed *p*-values

reported on the y-axis. These results show BNIP3 and NIX have significantly elevated phosphorylation events across all tested model systems. **f** Cartoon schematic of primary sequence of mouse BNIP3 (top) and NIX (bottom). Numbers indicate total amino acids in proteins. Select domains are highlighted: LC3 Interaction Region (LIR, green), Minimal Essential Region (MER, teal), BH3-only domain (BH3, purple), and transmembrane domain (TM, brown). Phosphorylation sites identified in our analysis are shown as pink lines. **g** List of identified phosphorylation sites on BNIP3 and NIX. Phosphorylated amino acid is listed at left of each column. Colored circles indicate the phosphorylation event is significantly upregulated (Student's *t* test, *p* < 0.05) in *Pptc7* knockout tissues relative to controls in perinatal heart (orange), perinatal liver (blue), mouse embryonic fibroblasts (green), or adult mouse liver tissue (purple). Source data are provided as a Source data file.

unbiased affinity enrichment-mass spectrometry, immunoprecipitation, yeast two-hybrid experiments, and through the direct dephosphorylation of these mitophagy receptors on mitochondria isolated from *Pptc7* knockout cells (Fig. 5). However, these data demonstrate only that PPTC7 *can* interact with BNIP3 and NIX in experiments in which they are proximal to one another. In cells, the matrix localization of PPTC7[39] should theoretically disallow the direct regulation of BNIP3 and NIX, which reside on the outer mitochondrial membrane[13,40]. However, a handful of observations suggest that the location and function of PPTC7 may be more complex than currently appreciated. First, our western blot in Fig. 2b reveals two sets of PPTC7 bands, one that migrates at the expected molecular weight (~33 kDa), and one that migrates substantially higher than this (~40 kDa). These data suggest that PPTC7 may reside in distinct populations in cells, although the molecular details underlying such differences in migration patterns remain to be elucidated. Second, the yeast ortholog of *Pptc7*, *PTC7*, is alternatively spliced to encode isoforms that localize either to the mitochondrial matrix or to the ER or nuclear envelope[41,42]. These data suggest that Ptc7p and PPTC7 may share evolutionarily conserved dual functions in distinct organellar or cellular locations. Finally, the potential dual localization of PPTC7 is further supported by the BioPlex interactome studies[43–45], which identify an interaction between PPTC7 and BNIP3. Notably, the BioPlex datasets also identify multiple interaction partners for PPTC7 in the mitochondrial matrix (e.g., DBT, RTN4IP1, TRMT2B, and TRMT61B) as well as on the outer mitochondrial membrane (e.g., RHOT1, RHOT2, HSDL1, MAOB, and TRABD), which could be consistent with dual localization of this phosphatase. We propose that these data, along with the data presented in this manuscript, support a role for PPTC7 in the direct dephosphorylation of BNIP3 and NIX. Whether these interactions are direct in cells and under what physiological contexts these interactions take place should be active areas of investigation in the future.

Despite the importance of BNIP3 and NIX in mediating excessive mitophagy in *Pptc7* knockout systems, the lack of full metabolic rescue in our TKO cells suggests that this phosphatase influences mitochondrial functions in ways distinct from its functional interaction with these mitophagy receptors. One possibility is that dysregulated phosphorylation of mitochondrial proteins, as seen in *Pptc7* KO systems, affects their mitophagic selectivity, as has recently been suggested in *S. cerevisiae*[46,47]. This could happen independently of BNIP3 and NIX, which are not conserved in budding yeast. Despite the lack of BNIP3 and NIX in yeast, Ptc7p promotes mitochondrial function[7,48], suggesting that PPTC7 may have ancestral roles in enabling mitochondrial metabolism distinct from its role in mammalian mitophagy. One such function may be the regulation of mitochondrial protein import via TIMM50. Phosphorylation at threonine 33 of TIMM50 is one of the four consistently upregulated phosphorylation events in *Pptc7* KO systems (Fig. 4c), and phosphorylation in a similar region of Tim50p (at serine 104) was identified in *ptc7Δ* yeast[7,14]. We previously demonstrated that phosphomimetic mutation of S104 in Tim50p decreases mitochondrial protein import and that a non-

phosphorylatable S104A Tim50p mutant partially rescues import defects in *ptc7Δ* yeast[14]. These data suggest that the dysregulated phosphorylation on TIMM50, or other candidate substrates identified in our analyses, likely contributes to the metabolic dysfunction seen in *Pptc7* KO systems. The TKO cells generated for this study will serve as a critical tool for disentangling these metabolic phenotypes in the absence of diminished mitochondrial protein content.

Collectively, our data demonstrate that loss of the mitochondrial phosphatase PPTC7 decreases mitochondrial protein content in isolated cells, in multiple tissue types, and across developmental contexts. This loss in mitochondrial protein content is associated with elevated phosphorylation on numerous mitochondrial proteins, including the mitophagy receptors BNIP3 and NIX. The decreases in mitochondrial protein content seen in *Pptc7* KO cells are largely rescued by genetic ablation of BNIP3 and NIX without rectifying the underlying metabolic issues. These data highlight the critical role of PPTC7 in maintaining mitochondrial homeostasis and further substantiate the importance of properly regulated protein phosphorylation within this organelle.

## Methods
### Generation of a conditional floxed Pptc7 mouse model
*Pptc7* floxed animals were generated at the Biotechnology Center at the University of Wisconsin-Madison. The *Easi*-CRISPR method[49] was used to generate two loxP sites flanking exon 3 of Pptc7 (i.e., a floxed allele). Animals were generated with a CRISPR approach as previously described (Niemi et al.[14]), with a key modification the addition of a long, single stranded oligonucleotide (i.e., a megamer) to facilitate homologous recombination. Briefly, one-cell fertilized C57BL/6J embryos were microinjected with CRISPR reagents (ctRNA.int12.2 (20 ng/µL) + ctRNA.int23.3 (20 ng/µL) + Cas9 IDT (50 ng/µL) + megamer (50 ng/µL)). Sequences of ctRNA.int12.2, ctRNA.int23.3 and the sequence of the resulting floxed allele is provided in Supplementary Table 1. Microinjected embryos were surgically transferred into pseudopregnant recipients. Pups were sequenced at weaning for each LoxP cassette. Founders were backcrossed to C57BL/6J mates and F1 pups analyzed by sequencing to evaluate germline transmission.

### Generation of an inducible global Pptc7 knockout mouse
*Pptc7* floxed animals were bred with UBC-Cre-ERT2 transgenic mice to generate animals a global, inducible Pptc7 knockout model. The UBC-Cre-ERT2 strain was acquired from Jackson laboratories (B6.Cg-Ndor1^Tg(UBC-cre/ERT2)1Ejb/1J, Strain #007001) and was bred to Pptc7^flox/flox animals. Pptc7 floxed animals were bred to homozygosity (i.e., Pptc7^flox/flox) while maintaining one copy of the UBC-Cre/ERT2 transgene (referred to as "Cre" hereafter) as experimental animals. Control animals include UBC-Cre/ERT2;Pptc7^+/+ animals and/or Pptc7^flox/flox animals carrying no copy of Cre transgene. Controls used in each experiment are clarified within the figure legends of the manuscript. Both male and female mice were used in our analyses. To induce

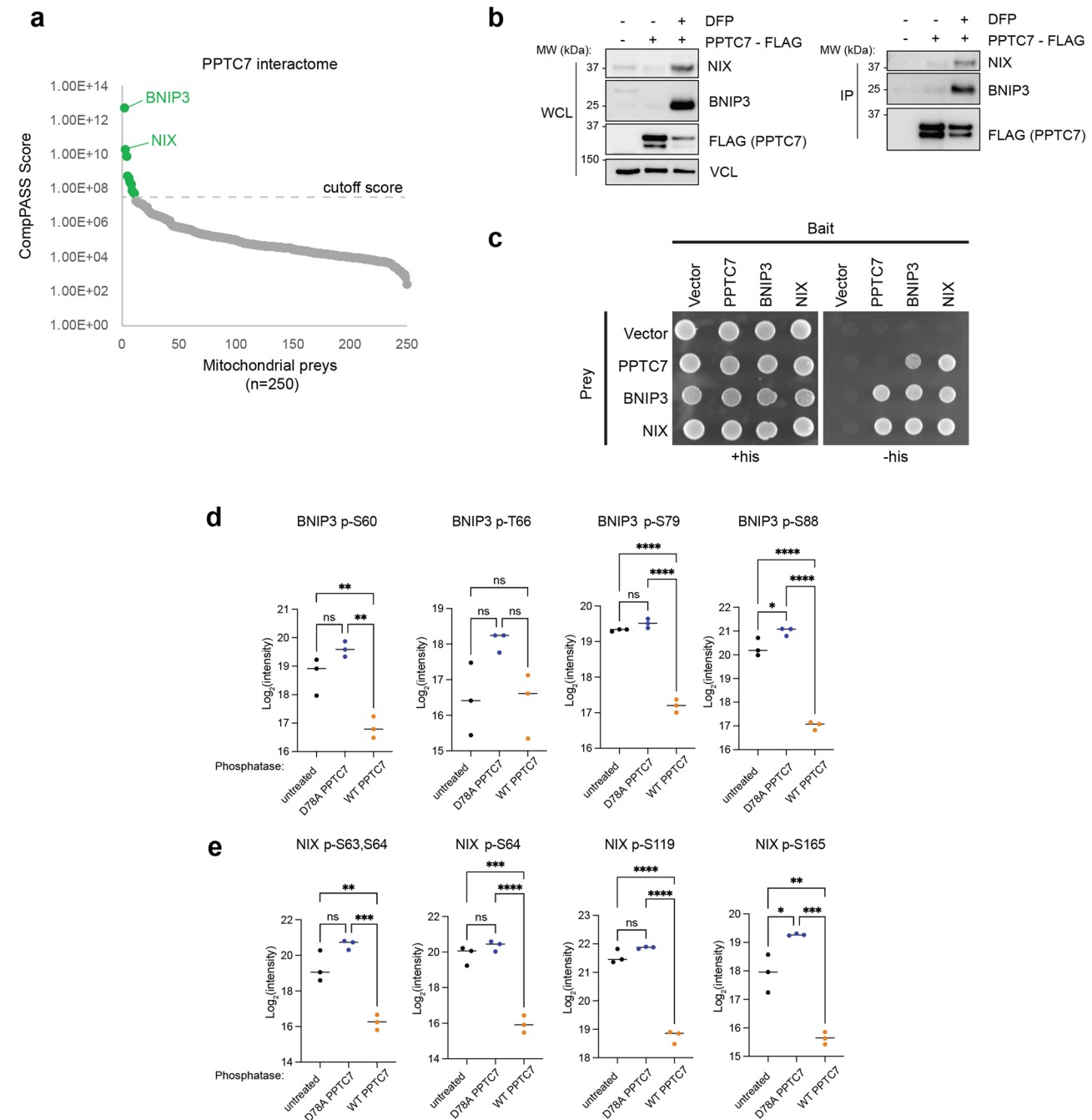

**Fig. 5 | PPTC7 interacts with and can dephosphorylate BNIP3 and NIX.**
**a** Mitochondrial proteins that interact with PPTC7 after affinity purification-mass spectrometry analysis as performed in ref. 22. **b** Immunoprecipitation (IP) of FLAG-PPTC7 overexpressed in U2OS cells (right panels) with endogenous BNIP3 and NIX in basal conditions (middle column) or after treatment with 1 mM DFP to induce BNIP3 and NIX expression (right column). Input shown as whole cell lysate (WCL) at right; anti-vinculin (VCL) shown as a load control. This experiment is representative of at least three independent experiments. **c** Yeast two-hybrid analysis of PPTC7, BNIP3, and NIX as reciprocal baits and preys. Yeast were spotted on permissive (+his, left) or selective (-his, right) plates. **d, e** Phosphoproteomic analysis of BNIP3 (**d**) or NIX (**e**) phosphopeptides isolated from *Pptc7* knockout mitochondrial fractions. Mitochondria were isolated from cells and split into n = 3 fractions for each treatment: untreated (black dots, left, n = 3 mitochondrial fractions), treated with catalytically inactive PPTC7 (D78A, blue dots, middle, n = 3 mitochondrial fractions) or active PPTC7 (orange dots, right, n = 3 mitochondrial fractions). The log2 intensity of each phosphopeptide is shown; each dot represents a biological replicate from one of the nine independently treated mitochondrial fractions outlined above. Lines represent median. Ordinary one-way ANOVA performed. ****p < 0.0001, ***p < 0.001, **p < 0.01, *p < 0.05, ns = not significant. p-values for WT v. D78A are as follows: BNIP3 S60 p = 0.001, BNIP3 T66 p = 0.1145, BNIP3 S79 p < 0.0001, BNIP3 S88 p < 0.0001, NIX S63,64 p = 0.0003, NIX S64 p < 0.0001, NIX S119 p < 0.0001, NIX S165 p = 0.0001. Source data are provided as a Source data file.

knockout, mice were fed tamoxifen-containing chow for seven consecutive days. Tamoxifen chow was purchased from Envigo (Teklad custom diet TD.130859), which is formulated for 400 mg of tamoxifen citrate per kilogram of diet. The diet was modified to promote more consistent feeding and genomic knockout as follows: tamoxifen

chow pellets were ground to a powder with a morter and pestle, as was standard chow (Formulab Diet 5008). The two diets were mixed 3:1 (e.g., 15 g of tamoxifen chow to 5 g of standard chow) and placed into a weigh boat. Water was added to the powder to create a "mash"; a fresh mash was given to each cage daily for the duration of the

treatment. After tamoxifen administration was complete, mice were transitioned back to a standard chow diet (Formulab Diet 5008) until the end of the experiment. Mice were housed on a 12-h light:dark cycle, and group housed by strain and sex under temperature- and humidity-controlled conditions and received ad libitum access to water and food. All animal work complied with ethical regulations for animal testing and research, and was done in accordance with IACUC approval by the College of Agricultural and Life Sciences (CALS) Animal Care and Use Committee at the University of Wisconin-Madison (protocol/animal welfare assurance #A3368-01) or at Washington University School of Medicine (animal welfare assurance D1600245).

### Genotyping analysis

Each model used in our studies was validated by genotyping analysis. Genotyping primers for the flox allele and Cre used for genotyping can be found in Supplementary Table 2. The flox primers produce a 667 bp amplicon for the floxed allele and a 599 bp amplicon for the wild type allele. Upon Cre-mediated excision, these primers produce a 167 bp amplicon reflecting a knockout allele. The Cre-specific primers produce a 480 bp amplicon for the Cre allele. Approximately 100 ng of genomic DNA isolated from tail tips served as a template for all genotyping reactions. *Pptc7* knockout MEFs were generated from our previously published CRISPR knockout model and genotyped as previously described[14].

### Generation and validation of mouse embryonic fibroblasts

Mouse embryonic fibroblasts were generated from a cross of Pptc7[+/-] heterozygous mice. A timed mating was performed and E0.5 was determined through the presence of a vaginal plug. The pregnant female was sacrificed at E14.5 by $CO_2$ asphyxiation. 11 embryos were removed, decapitated, and internal organs removed. Remaining tissue was finely minced, suspended in 0.25% trypsin-EDTA, and incubated at 37 °C for 30 min in a $CO_2$-controlled tissue culture incubator. After trypsinization, tissues were pipetted to a single cell suspension, filtered through a nylon mesh cell strainer (Thermo-Fisher #08-771-1), and plated into individual 60 mm$^2$ dishes. Cells were cultured in DMEM (high glucose, no pyruvate, Thermo-Fisher catalog #11965-092) supplemented with 10% heat inactivated fetal bovine serum (FBS) and 1× penicillin/streptomycin. Cells were grown in a temperature-controlled $CO_2$ incubator at 37 °C and 5% $CO_2$. For the first 3 passages, media was supplemented with gentamicin (10 μg/mL) to reduce risks of contamination. Each cell line was genotyped using leftover tissue from each embryo from which genomic DNA was isolated. gDNA isolation and genotyping was done as previously described[14]. The sex of the cells was not determined as male/female embryos are morphologically indistinguishable at E14.5. Immortalized MEFs were generated using plasmid encoding SV40 Large T antigen as follows: MEFs were split into 6 well plates to ~80% confluence and transfected with 2 μg SV40 1: pBSSVD2005 (a gift from David Ron; Addgene plasmid #21826) using FuGENE HD according to manufacturer's protocol. MEFs were split 1:10 for 5 consecutive passages before considered immortalized. Immortalized MEFs were cultured identically to primary MEFs (see above) and were used to a maximum of 15 passages for all experiments. Cellular experiments in this paper were performed with both primary (e.g., Seahorse assays) and immortalized (e.g., phosphoproteomics, proteomics, and BioLog assays) MEF clones.

### Proteomic and phosphoproteomic analysis

TMT-based proteomic and phosphorproteomic experiments were performed on mouse liver tissue and mouse embryonic fibroblasts, Supplementary Data 1 and 2, respectively. Label-free proteomics analysis was performed on *Pptc7/Bnip3/Bnip3l* TKO samples, Supplementary Data 3. Details for the generation of these datasets are as follows:

### TMT phosphoproteomics

**Tissue lysis and protein digestion.** Cells or mouse liver tissue were resuspended in 1 mL 6 M guanidine hydrochloride, 100 mM Tris, pH 8 and probe-sonicated using a Misonix XL-2000 Series Ultrasonic Liquid Processor. The sonication regime, repeated twice, was the following: the probe was set to output 11 watts for 10 s with an off-time of 30 s. The protein concentrations of the homogenized tissue samples were determined using the BCA Protein Assay Kit (Thermo Pierce). Sample volumes corresponding to 400 μg of protein were set aside for further processing. Methanol was then added to each sample so that the final volume consisted of 90% methanol. The samples were then centrifuged at $9000 \times g$ for 5 min, and the supernatant was decanted. The protein pellet was then resuspended in 200 μL of lysis buffer containing 8 M urea, 10 mM tris (2-carboxyethyl) phosphine, 40 mM chloroacetamide, and 100 mM Tris, pH 8. 8 μg of Lysyl Endopeptidase (FUJIFILM Wako Chemicals) was added and incubated for 4 h at room temperature. The samples were then diluted with 100 mM tris, pH 8 to achieve a final urea concentration of less than 2 M. 8 μg of trypsin (Promega) was added, and the samples incubated for 10 h at room temperature. Trypsin enzyme activity was then quenched by addition of 120 μL of 10% trifluoroacetic acid (TFA), and the samples were centrifuged at $9000 \times g$ for 5 min. Peptides were desalted with Strata-X reversed phase solid phase extraction cartridges (Phenomenex).

**TMT labeling.** Desalted peptides were resuspended in 100 μL 200 mM triethylammonium bicarbonate buffer. Each TMTpro 16-Plex channel had been pre-aliquoted in 500 μg quantities was resuspended in 20 μL of acetonitrile (ACN). Each sample was added to distinct TMT channels and vortexed for 4 h at room temperature. The TMT labeling reaction was quenched with 0.64 μL of 50% hydroxylamine and 100 μg of labeled peptides per sample were combined. The combined sample was dried and desalted with Strata-X reversed phase solid phase extraction cartridges.

**Phosphopeptide enrichment.** Desalted TMT-labeled peptides were resuspended in 1 mL 8% can in 6% TFA. 104 μL of magnetic titanium dioxide beads (ReSyn Biosciences) that were washed with 80% ACN 6% TFA three times. The resuspended sample was added to the washed beads and vortexed for 1 h at room temperature. The bead-bound phosphorylated peptides were then sequentially washed three times with 1 mL of 80% ACN/6% TFA, once with 80% ACN, once with 80% ACN in 0.5 M glycolic acid, then three times with 80% ACN. The phosphorylated peptides were eluted from the beads with two washes of 300 μL 50% ACN in 1% ammonium hydroxide. Dried phosphorylated peptides were resuspended in 0.1% TFA and desalted using Strata-X reversed phase solid phase extraction cartridges.

**Offline fractionation.** The TMT-labeled phosphopeptides were loaded onto a 4.6 mm inner diameter 150 mm outer diameter BEH C18 column (Waters Corp) for offline fractionation into 16 fractions. The peptides were separated using a high pH reversed-phase gradient (10 mM ammonium formate Mobile Phase A, 10 mM ammonium formate in 80% MeOH Mobile Phase B) on an Agilent 1260 Infinity Binary LC equipped with an automatic fraction collector. The effective separation gradient started at 20% Mobile Phase B and increased to 60% Mobile Phase B in 6 min, flowing at a rate of 800 μL/min. The fractions were then concatenated into 8 fractions, combining fractions 1 and 9, 2 and 10, 3 and 11, 4 and 12, 5 and 13, 6 and 14, 7 and 15, and 8 and 16. The combined fractions were transferred to 2 mL Starstedt micro tubes for rapid evaporation under vacuum with the SpeedVac Vacuum Concentrator Kit (Thermo).

**LC-MS acquisition.** Each fraction was resuspended in 0.2% formic acid. 1 μg of peptide was loaded onto a 75 μm inner diameter × 360 μm outer diameter column (New Objective) packed in-house 1 to 30 cm

with 1.7 micron BEH C18 particles (Waters Corp). The peptides were chromatographically separated over a 120-min reverse phase gradient (0.2% formic acid mobile phase A, 0.2% formic acid/80% ACN mobile phase B) using a Dionex UltiMate 3000 nano-HPLC (Thermo) at a column temperature of 50 °C and a flow rate of 335 nL/min. Eluting peptides were sprayed into an Orbitrap Eclipse using a Nanospray Flex ionization source (Thermo) at 2 kV. MS1 scans were recorded in the Orbitrap using a resolving power of 60,000, requiring an AGC target of 1e6 or a maximum injection time of 50 ms. MS1 scans included precursor ions ranging from 300-1350 $m/z$. The most intense ions were selected for HCD fragmentation (normalized collision energy 35%) and subsequent MS2 analysis until the 1 s duty cycle lapsed. Dynamic exclusion was set to 40 s and the MS2 isolation width was set to 1.5 Th. Only precursors of charge states 2–6 were selected. MS2 spectra were analyzed in the Orbitrap using a resolving power of 60,000 requiring an AGC target of 1e5 or a maximum inject time of 118 ms. MS2 scans included fragment ions ranging from 100 to 2000 $m/z$.

**Data processing.** Thermo RAW files were processed with MaxQuant (version 1.5.2.8)2, implementing the Andromeda3 algorithm to perform database searching of MS2 spectra against a database of canonical proteins and isoforms (Uniprot, *Mus musculus*, 2018). TMT labels were set as fixed modifications on N termini and lysine residues. Carbamidomethylation of cysteine was set as a fixed modification. Variable modifications included acetylation of the N terminus, phosphorylation of serine, threonine, and tyrosine, and oxidation of methionine. Peptide and protein matches were filtered to 1% FDR. Phosphorylation sites were considered localized if they delivered MaxQuant localization scores >0.75. The TMT reporter ion intensities of the phosphorylated peptides were median-normalized to account for variability introduced from sample-handling. To this end, the median of the reporter ion intensities for each sample was calculated. To obtain the correction factor for each phosphorylated peptide intensity, the average of the medians was divided by each of the respective sample medians, rendering a correction factor for each channel. This channel-specific correction factor was then multiplied to each of the reporter ion intensities within that channel.

### Label-free proteomics

**Protein extraction and digestion.** Mouse embryonic fibroblast cells were resuspended in 1 mL 6 M guanidine hydrochloride, 100 mM Tris, pH 8 and sonicated using a Qsonica Q700 temperature-controlled sonicator. The sonication regime cycled 10 times applying an ultrasonic wave of amplitude of 35 for 20 s with a 10 s off time. The protein concentrations of the lysate were then determined using the Pierce BCA Protein Assay Kit. Methanol was then added to volumes corresponding to 200 μg for each sample, so that the final volume consisted of 90% methanol for protein extraction. The samples were then centrifuged at 9,000 g for 5 min and the supernatant was decanted. The protein pellet was then resuspended in 100 μL of lysis buffer containing 8 M urea, 10 mM Tris (2-carboxyethyl) phosphine, 40 mM chloroacetamide, and 100 mM Tris, pH 8. 4 μg of Lysyl Endopeptidase was added and incubated for 4 h at room temperature. The samples were then diluted with 100 mM tris, pH 8 to achieve a 2 M urea concentration. 4 μg of trypsin was added and the samples incubated for 10 h at room temperature. Trypsin enzyme activity was then quenched by adding 60 μL of 10% TFA and the samples were centrifuged at 9000 × $g$ for 5 min. Peptides were desalted with Strata-X reversed phase solid phase extraction cartridges.

**LC-MS acquisition method.** Each sample was resuspended in 0.2% formic acid. 1.5 μg of peptide was loaded onto a 75 um inner diameter × 360 um outer diameter column (CoAnn Technologies) packed in-house to 30 cm with 1.7 micron BEH C18 particles (Waters Corp). The peptides were chromatographically separated over a 90-min reverse

phase gradient (0.2% formic acid mobile phase A, 0.2% formic acid/80% ACN mobile phase B) using a Vanquish Neo UHPLC (Thermo) at a column temperature of 50 °C and a flow rate of 320 nL/min. Eluting peptides were sprayed into an Orbitrap Eclipse using a Nanospray Flex ionization source (Thermo) at 2.2 kV. MS1 scans were recorded in the Orbitrap using a resolving power of 240,000, requiring an AGC target of 8e5 or a maximum injection time of 50 ms. MS1 scans included precursor ions ranging from 350-2000 m/z. The most intense ions were selected for HCD fragmentation (normalized collision energy of 25%) and subsequent MS2 analysis until the 1 s duty cycle lapsed. Dynamic exclusion was set to 10 s and the quadrupole isolation width was set to 0.5 Th. Only precursor ions with charge states of 2–5 were selected. MS2 spectra were analyzed in the ion trap on the "turbo" resolution setting, requiring an AGC target of 2e4 or a maximum inject time of 14 ms. MS2 scans included fragment ions ranging from 150 to 1350 $m/z$.

**Data processing.** Thermo RAW files were processed with MaxQuant (version 1.5.2.8)2, implementing the Andromeda3 algorithm to perform database searching of MS2 spectra against a database of canonical proteins and isoforms (Uniprot, *Mus musculus*). Carbamidomethylation was set as a fixed modification on cysteines. Variable modifications included N terminus acetylation and methionine oxidation. Peptide and protein matches were filtered to 1% FDR. Fast LFQ was toggled off and a LFQ minimum ratio count was set to 1. Match between runs (MBR) was toggled on. All other parameters were set to default MaxQuant settings.

### Phosphoproteomics on phosphatase-treated isolated mitochondria

*Pptc7[-/-]* mouse embryonic fibroblasts were grown as detailed in *"Generation and validation of mouse embryonic fibroblasts"*. Crude mitochondria were isolated as previously described[50] with some modifications. Briefly, cells were scraped into dPBS, placed on ice, and pelleted at 600 × $g$ at 4 °C for 10 min. Cells were resuspended in ice-cold IB$_c$ (0.1 M Tris-MOPS, pH 7.4, 0.1 M EGTA). Cells were transferred to a prechilled 7 mL dounce homogenizer on ice, and cells were homogenized using a tight pestle with 30–35 strokes. Homogenized cells were spun at 600 × $g$ at 4 °C for 10 min to pellet nuclei and unbroken cells. The supernatant was transferred to a new, prechilled tube and spun at 7000 × $g$ for 10 min at 4 °C. The pellet contained crude mitochondria, which were resuspended in IB$_c$ and quantified using the Pierce 660 nm protein assay. ~500 μg mitochondria per condition were diluted into IB$_c$ containing 10 mM MnCl$_2$ and ~1 μg of PPTC7 or PPTC7 D78A (or vehicle only for untreated conditions). Phosphatase assays were incubated at room temperature for 60 min before being snap frozen and stored at −80 °C until processing for phosphoproteomics.

**Sample preparation.** Crude mitochondrial protein was precipitated in 90% LC-MS grade MeOH and centrifuged at 16,500 × $g$ at 4 °C for 10 min. Pellets were solubilized in buffer containing 8 M Urea and 100 mM Tris by sonication. Samples were centrifuged again (16,500 × $g$ at 4 °C for 10 min), and supernatant protein concentration was quantified using BCA assay. Total protein was normalized to 400 μg per sample prior to reduction and alkylation with 10 mM TCEP and 40 mM 2-CAA, respectively. Urea concentration was reduced to <2 M with 100 mM Tris prior to tryptic digestion (1:25) overnight shaking at 37 °C. Digested peptides were desalted using StrataX polymeric reverse phase resin (Phenomenex), and fully dried before further processing. 5% of total peptides were retained as a proteomics sample. The remaining peptides were subjected to Sequential enrichment using Metal Oxide Affinity Chromatography (SMOAC) using TiO$_2$ and Fe$^{3+}$ enrichment columns, and dried to completion. Proteomic, TiO$_2$, and Fe$^{3+}$ samples were resuspended in 0.2% formic acid (FA).

**LC-MS/MS analysis.** 1 μg of peptides for proteomics analysis, and 4 μL of phosphoenriched samples were injected to a Thermo Orbitrap Exploris 240 following LC separation on an EASY-spray HPLC column (Thermo, ES902) using an UltiMate 3000 UHPLC system. Proteomics samples were separated using buffer A (0.2% formic acid) and buffer B (0.2% FA in 80% acetonitrile) on a 120 min gradient (3–60 min, 4–20% B, 60–100 min, 20–45% B, then washed and re-equilibrated). Phosphoproteomic samples were separated on a 120 min gradient (5–100 min, 8–40% B, then washed and re-equilibrated). Eluted peptides entered the mass spectrometer following positive mode ESI. For proteomics samples, MS1 scans were performed in the orbitrap (120,000 resolution, normalized AGC target 300%, RF Lens 80%, maximum ion injection time of 50 ms, with a 1.0e5 intensity threshold). MS2 analysis of HCD-generated (75% collision energies) fragmented ions were performed in the orbitrap (30,000, normalized AGC target 75%, with automatic maximum injection time). For phosphoproteomic samples, MS1 scans were performed in the orbitrap (120,000 resolution, normalized AGC target 300%, RF Lens 70%, maximum ion injections time 100 ms, with a 5.0e3 intensity threshold). MS2 analysis of HCD-generated (25% collision energies) fragmented ions were performed in the orbitrap (30,000, normalized AGC target 25%, with automatic maximum injection time). Monoisotopic precursor selection and dynamic exclusion (proteomics 60 s, phosphoproteomics 30 s) were enabled.

**MS data analysis.** Thermo raw files were searched with Proteome Discoverer 2.5 using Sequest HT with the uniprot *Mus musculus* fasta file (v2023-03-01). Searches were performed with a precursor mass tolerance of 10 ppm. Carbamidomethylation (+57.021 Da (C)) was included as a static modification, and N-terminal Acetyl (+42.011 Da), Met-loss (−131.040 Da (M)), and Met-loss+Acetyl (−89.030 Da (M)) were included as dynamic modifications in all searches. Phosphorylation (+79.966 Da (S, T, Y)) was included as a dynamic modification for all phosphoenrichment samples. The IMP-ptmRS node was used for scoring post-translational modifications, with a cut off set to >0.75. Search results were filtered to a peptide FDR of 1%. Phosphopeptide intensities were normalized to total reporter ion intensity. Imputation was performed by sampling values from the lowest 5% of the normal distribution within groups. Associated *p*-values were calculated using Student's T-test assuming equal variance.

## Quantification of ketones, triacylglycerols, and cholesterol

Ketone bodies were quantified as previously described[14]. Briefly, serum from mice fasted overnight (i.e., between 16 and 18 h) was collected and stored at −80 °C until the assay was performed. Serum was thawed on ice, diluted 1:10, and ketone bodies were quantified using the Wako Autokit Total Ketone Bodies Assay per the manufacturer's directions, with the modification that measurements were made kinetically over a 30-min period rather than an endpoint measurement. A standard curve, made from Wako's Calibrator 300, was simultaneously run and total ketone bodies in each sample were calculated according to this standard. Quantification of total triglycerides and total cholesterol in plasma were determined using the Triglyceride Infinity Kit and the Cholesterol Infinity Kit (both from Thermo Scientific). Briefly, in a 96 well plate format, 100 μL of each reagent was aliquoted and 1 μL of plasma sample or standard was added and mixed. Reactions were incubated for 30 min at room temperature. Total triglycerides were determined by readings at an absorbance of 540 nm, while total cholesterol was read at an absorbance of 490 nm. Sample concentrations were calculated from a standard curve correcting for blanks and a secondary wavelength at 660 nm. The same kits were used to quantify total triglycerides and total cholesterol in liver tissue. For sample preparation, 30-50 mg of fresh weight liver tissue was homogenized in 1.5 mL of chloroform: methanol (2:1 v/v). Samples were centrifuged at 14,000 × *g* for 10 min at 4 °C. An aliquot was evaporated in a 1.5 mL microcentrifuge tube and 100 μL of reagent was added to the dried sample followed by 30-min room temperature incubation. The developed reagent was then transferred to a 96 well plate with standards to determine the concentration. Final results were calculated by normalizing for the initial fresh weight of tissue.

## Metabolic assays

For the Seahorse assays in Fig. 2, primary (i.e., not immortalized) *Pptc7* knockout MEFs and matched wild-type MEFs were plated into Seahorse XFe96 plates at 12,500 cells/well. For the Seahorse assay in Fig. 3, immortalized wild-type, *Pptc7* knockout, and *Pptc7/Bnip3/Bnip3l* triple knockout cells were plated into XFe96 plates at 15,000 cells/well. Cells were allowed to adhere overnight in standard DMEM media (25 mM glucose, 2 mM glutamine, 10% heat inactivated FBS, 1× penicillin/ streptomycin). One hour before the Seahorse experiment, DMEM was aspirated, cells were washed with PBS, and were swapped into Seahorse XF Base Medium (i.e., unbuffered DMEM containing 25 mM glucose, 2 mM glutamine with no FBS) and allowed to equilibrate at 37 °C in a non-CO₂ incubator before initiating the experiment. While cells were equilibrating, a pre-hydrated Seahorse cartridge was loaded with oligomycin, FCCP, and rotenone/antimycin A prepared from a XF Cell Mito Stress Test Kit, which were reconstituted per the manufacturer's directions. Compounds were injected into 180 μL assay medium in subsequent injections for final concentrations of 1.0 μM oligomycin, 1.0 μM FCCP, and 0.5 μM rotenone/antimycin A. For pyruvate/malate and succinate/rotenone experiments, primary *Pptc7* knockout and wild-type MEFs were permeabilized using XF PMP per manufacturer's instructions. MEFs were plated in a 96 well plate as described above. Plating DMEM was aspirated, and cells were washed with 1× MAS (220 mM mannitol, 70 mM sucrose, 10 mM KH₂PO₄, 5 mM MgCl₂, 2 mM HEPES, 1 mM EGTA, and 0.2% w/v fatty acid free BSA). To initiate the experiment, cells were swapped into 1x MAS containing 1 nM XF PMP (to permeabilize the cells) and either 1 mM malate and 10 mM pyruvate or 10 mM succinate and 2 μM rotenone. Permeabilized cells were subjected to the XF Cell Mito Stress Test Kit as described above. For the fatty acid oxidation Seahorse experiment, palmitate-BSA and BSA were purchased from Agilent to test the responsiveness of wild type and *Pptc7* knockout MEFs to fatty acids. Briefly, 15,000 immortalized mouse embryonic fibroblasts were plated into XFe96 plates in DMEM and allowed to adhere overnight. Twenty-four hours before starting the assay, growth media was replaced with substate-limited medium (i.e., DMEM, 0.5 mM glucose, 1 mM glutamine, 0.5 mM carnitine, and 0.5% FBS). 45 min before starting the assay, cells were washed with PBS and media was replaced with FAO assay medium (KHB – 111 mM NaCl, 4.7 mM KCl, 1.25 mM CaCl₂, 2 mM MgSO₄, 1.2 mM NaH₂PO₄) supplemented with 2.5 mM glucose, 0.5 mM carnitine, and 5 mM HEPES, adjusted to pH 7.4. 15 min prior to starting the assay, 4 μM etomoxir or vehicle (DMSO) was added to select wells. Palmitate-BSA or BSA alone were injected to wells after three measurements of basal respiration. All data were collected on an XFe96 Seahorse Flux analyzer and analyzed on its associated Seahorse XF software. To perform the BioLog experiment, a BioLog MitoPlate S-1 Mitochondrial Substrate Metabolism assay plate was activated by adding "assay mix" (BioLog MAS, Redox Dye MC, saponin, and water) to each well and incubating for 1 h at 37 °C to dissolve substrates. Immortalized *Pptc7* knockout MEFs and matched wild type MEFs were trypsinized, counted to a final cell number of 20,000 cells per well, and resuspended in 3 mL of 1× BioLog MAS (Biolog catalog #72303). The redox dye color formation (OD590) within the plate was tracked on an OmniLog instrument for 4 h. Data were collected on OmniLog software and analyzed by calculating the slope of redox dye formation as reported over this 4-h timeframe.

## Citrate synthase assays

Citrate synthase assays were performed as previously described[7,14]. Briefly, liver tissue was homogenized in Lysis Buffer A (LBA; 50 mM Tris-HCl, pH 7.4, 40 mM NaCl, 1 mM EDTA, and 0.5% (v/v) Triton X-100)

supplemented with 1× protease inhibitor cocktail (0.5 μg/mL pepstatin A, chymostatin, antipain, leupeptin, and aprotinin) and 1× phosphatase inhibitor cocktail (0.5 mM imidazole, 0.25 mM sodium fluoride, 0.3 mM sodium molybdate, 0.25 mM sodium orthovanadate, and 1 mM sodium tartrate). Lysates were clarified, BCA normalized, and 5 μg of whole cell lysate was used for each reaction. 200 μL reactions were assembled into a single well of a 96 well plate containing the following (final concentrations): 100 mM Tris, pH 7.4, 300 μM acetyl CoA, 100 μM DTNB (5,5'-dithio-bis-[2-nitrobenzoic acid]). 10 μL of 10 mM oxaloacetic acid (OAA) was added per well (final [c] = 500 μM) to initiate enzyme assay. Absorbance at 412 nm (A412) was measured every minute for 30 min on a Cytation 3 plate reader (BioTek).

### Western blotting and antibodies
Western blotting was performed as previously described[14]. Antibodies used in this study include: anti-PPTC7 (Novus Biologicals, catalog #NBP1-90654, 1:1000 dilution, 48 h incubation), anti-Citrate synthase (Cell Signaling Technologies, catalog #14309, 1:1000 dilution, overnight incubation), anti-beta actin (Abcam, catalog # ab170325, 1:1000 dilution, overnight incubation), anti-BNIP3 (rodent specific antibody, Cell Signaling Technologies, catalog #3769, 1:1000 dilution, 48 h incubation), anti-NIX (Cell Signaling Technologies, catalog #12396, 1:1000 dilution, overnight incubation), and anti-LC3B (Cell Signaling Technologies, catalog #3868, 1:1000 dilution, overnight incubation). For the immunoprecipitations, the following antibodies were used: anti-BNIP3 (Abcam, clone EPR4034, catalog #ab109362, WB 1:1000), anti-NIX (Cell Signaling Technology, clone D4R4B, catalog #12396, WB 1:1000); anti-FLAG (Sigma-Aldrich, catalog #SAB4301135, WB 1:1000), and anti-vinculin (SCBT, clone G-11; sc-55465, WB 1:1000).

### mtDNA analysis
Genomic DNA (i.e., gDNA) was isolated from mouse liver tissue using the DNeasy Blood & Tissue Kit (Qiagen) according to manufacturer's instructions. DNA was quantified and normalized to 100 ng input per reaction. Primers to analyze nuclear and mitochondrial DNA markers can be found in Supplementary Table 3. Quantitative real time PCR was performed on a QuantStudio 6 Real-Time PCR system with data was collected on QuantStudio Real-Time PCR v1.2 software. Relative quantitation was performed using the ΔΔCt method.

### Generation of Pptc7/Bnip3/Nix triple knockout (TKO) cells
A single clone of *Pptc7* knockout MEFs were used to generate *Pptc7/Bnip3/Bnip3l* triple knockout (TKO) cells. The TKO MEFs were created by the Genome Engineering & Stem Cell Center (GESC@MGI) at Washington University in St. Louis. Briefly, two gRNAs were designed to target each 5' and 3' of exon of interest, introduce double strain break and delete target exon. Synthetic gRNAs were purchased from IDT (sequences are provided in Supplementary Table 4), complexed with Cas9 recombinant protein and transfected into MEFs. The transfected cells were then single cell sorted into 96-well plates. Single cell clones were identified using deletion PCR and inside PCR to analyze the target site region for complete deletion of the respective gene (primers sequence and PCR products size are reported in Supplementary Table 4).

### Mitophagy and autophagy assays
The mitochondrial keima (mt-Keima) assay was described previously[19]. Live-cell images were acquired using a Leica DMi8 SP8 Inverted Confocal microscope, equipped with a 63x Plan Apochromatic objective and environmental chamber (5% CO$_2$, 37 °C). Images were quantified with Image J/FIJI software. Single cells were segregated by generating regions of interest (ROI). Selected ROIs were cropped and split into separated channels for 561 nm and 458 nm, followed by threshold processing. The fluorescence intensity of mt-Keima at 561 nm (lysosomal signal) and mt-Keima at 458 nm (mitochondrial signal) was

measured for each ROI and the ratio (561 nm/458 nm) was calculated. Three biological replicates were performed, with >50 cells analyzed per condition for each replicate. Individual ratios are represented as gray dots and the mean ratios from each biological replicate. For the flow cytometry experiments, cells were harvested by trypsinization and were resuspended in FluoroBrite media with 0.8% heat-inactivated FBS immediately prior to flow cytometry. Cells were analyzed on a LSR-Fortessa (BD Biosciences; Franklin Lakes, NJ, USA) flow cytometer using BDFACSDiva software (version 9.0). FSC (488 nm), SSC (excitation 488 nm, emission 488), QDot 605 (excitation 405 nm, emission 610), and PE-TexasRed (excitation 585 nm, emission 610) laser filters were used. For all experiments, wild-type MEFs (i.e., not expressing mtKeima) were used as a negative control to calibrate the laser intensities for Qdot 605 and PE-TexasRed, which were kept constant throughout experiments. Cells were gated to select for live cells, single cells, and mtKeima positive cells sequentially. Flow cytometry data were processed with FlowJo (version 10.2.2), and the high mitophagy gate was drawn during analysis. To determine autophagic flux in wild type and *Pptc7* knockout fibroblasts, cells were treated with the 50 nM bafilomycin A for an acute timecourse. Cells were then washed in PBS, lysed in RIPA buffer, and processed for western blotting for LC3B. 30 μg total lysate was loaded onto a 4-20% gradient gel, blotted to nitrocellulose membrane, and probed for LC3B (Cell Signaling Technologies, 1:1000). Bands were visualized on a Li-COR Odyssey and analyzed using the associated ImageStudio software.

### Yeast two hybrid analysis
Yeast two-hybrid analysis was performed as previously described[37]. Briefly, the Matchmaker Gold Yeast Two-Hybrid System (Takara) was used. Human BNIP3 and human NIX were cloned into pGBKT7 and pGADT7 as previously described[37]; for this study, full length human PPTC7 was cloned into pGBKT7 and pGADT7 using standard restriction-based cloning using the BamHI and XhoI restriction sites for pGADT7 and BamHI and NotI restriction sites for pGBKT7. Plasmids were verified by Sanger sequencing. Y2H Gold and Y187 yeast strains were transformed with bait (pGBKT7 plasmids and derivatives) and prey (pGADT7 plasmids and derivatives), respectively, by lithium acetate transformation. Haploid bait- and prey-expressing strains were mated on YPD plates (1% yeast extract, 2% peptone, and 2% glucose) for 24 h and diploids were subsequently selected on synthetic dextrose (SD; 0.7% yeast nitrogen base, 2% glucose, and amino acids) -leu-trp plates. Cells were grown to exponential phase in SD-leu-trp media, normalized to 0.5 OD 600 per mL, and cells were spotted on SD-leu-trp (permissive) and SD-leu-trp-his (selection) plates. Plates were then incubated at 30 °C prior to analysis.

### Co-immunoprecipitation assays
U2OS cells were transfected with PPTC7-FLAG using Lipofectamine 2000 (Thermo Fisher). Where indicated, cells were treated with 1 mM diferoprone (DFP). Cells were lysed on ice for 30 min in Tris-Triton lysis buffer (50 mM Tris-Cl pH 7.5, 150 mM NaCl, 10% glycerol, 1 mM EDTA, 1 mM EGTA, 5 mM MgCl2, 1 mM β-glycerophosphate, and 1% Triton) containing protease inhibitor cocktail (Rowe Scientific; CP2778) and PhosSTOP EASYpack Phosphatase Inhibitor Cocktail (Roche; 4906837001). Cell lysates were collected by centrifugation at 18,800 × *g* for 10 min at 4 °C. To affinity-precipitate exogenously expressed FLAG-tagged PPTC7, cell lysates were incubated in a rotating incubator for 1 h at 4 °C with anti-FLAG affinity gel (Sigma; A2220). The immunoprecipitates were washed with Tris-Triton lysis buffer 5 times prior to elution with Bolt™ LDS Sample Buffer, electrophoresis on Bolt 4–12% gradient gels, and Western blotting.

### Data and statistical analysis and software
Statistical analysis was carried out using either Microsoft Excel or Prism using the statistical tests listed in the associated figure legends. Volcano plots and linear regression plots were generated using

Microsoft Excel. Bar graphs, box plots, and dot plot distributions (e.g., Fig. 2d) were generated using Prism 9 (GraphPad). All graphs were imported into Adobe Illustrator for final adjustments. Seahorse figures were generated using Wave software, Excel, or Prism. FACS plots were generated using FlowJo. The GO analysis graph plotted in Fig. 1f was generated at: http://bioinformatics.sdstate.edu/go/. BioRender was used for some graphics.

## Reporting summary

Further information on research design is available in the Nature Portfolio Reporting Summary linked to this article.

## Data availability

The raw data for the proteomics and phosphoproteomics datasets in this study have been deposited to the MassIVE database under the accession number MSV000091194. Source data are provided with this paper.

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

## Acknowledgements

We would like to thank members of the Pagliarini lab and the Niemi lab for discussions and critical evaluation of this work. We would like to thank Sangeeta Adak and the Diabetes Models Phenotyping Core (P30DK020579) for assistance in the Biochemical analysis of tissue and serum. This work was supported by R01DK098672 (to D.J.P.), P41GM108538 (National Center for Quantitative Biology of Complex Systems, to D.J.P. and J.J.C.), R35GM151130 (to N.M.N.), the Diabetes Research Center at Washington University (P30DK020579 Pilot and Feasibility funds to N.M.N.), T32G002760 (Genomic Sciences Training Program training grant support to L.R.S.), R35GM137894 (to J.R.F.), and funds from the BJC Investigators Program (to D.J.P.). L.W. was supported by the MilliporeSigma Predoctoral Fellowship in Honor of Dr. Gerty T. Cori at Washington University. Support for this research was provided by the University of Wisconsin–Madison, Department of Biochemistry and Office of the Vice Chancellor for Research and Graduate Education with funding from the Wisconsin Alumni Research Foundation (to M.P.K.), and by NIH grants, R01DK101573, R01DK102948, and RC2DK125961 (to A.D.A.). J.K.P. was supported by Australian National Health and Medical Research Council grants (APP1183915 and APP2019993), a Brain Foun-dation Research grant (2020), and an Australian Research Council Future Fellowship (FT180100172). J.K.P and K.-L.K. were supported by a Mito Foundation Incubator Grant (2022) and Mito Foundation Ph.D. scholarship, respectively. We would like to thank Kathy Krentz and C. Dustin Rubenstein from the Animal Models and Genome Editing Cores at the University of Wisconsin-Madison for generating of the *Pptc7* floxed model. The generation of the *Pptc7* floxed model was supported by University of Wisconsin Carbone Cancer Center Support Grant (P30CA014520). We thank the Genome Engineering and iPSC Center (GESC@MGI) at the Washington University in St. Louis for creating the *Pptc7/Bnip3/Bnip3l* triple knockout (TKO) MEFs lines. We would like to acknowledge the Washington University Diabetes Research Center (P30DK020579) for providing access to a Seahorse instrument used to generate data in this manuscript.

## Author contributions

N.M.N. and D.J.P. wrote the manuscript. N.M.N. and D.J.P. conceived the overall project and its design. N.M.N., L.R.S., L.K.M., C.E.B., A.J.S., E.S., D.J.P., and J.J.C. generated and prepared samples, performed pro-teomics or phosphoproteomics, or assisted with data analysis and interpretation of proteomics and/or phosphoproteomics data. L.W., K.-L.K., and J.K.P. generated reagents, performed experiments, and ana-lyzed data regarding the mitophagy experiments. N.M.N., M.F., and E.H.R. performed experiments and analyzed data associated with cel-lular metabolic experiments. N.M.N., K.L.S., M.P.K., A.D.A., and D.J.P. assisted in generating and providing husbandry to the animals in this study, as well as advising on experimental design and data interpreta-tion. N.M.N., A.J.S., K.-L.K., O.M.C., J.R.F., J.K.P., and D.J.P. designed and performed experiments or analyzed data regarding the BNIP3/NIX-PPTC7 interactions. All authors reviewed and edited the manuscript.

## Competing interests

The authors declare the following competing interests: J.J.C. is a con-sultant for Thermo Fisher Scientific, 908 Devices, and Seer. The remaining authors declare no competing interests.

## Additional information

[1]Morgridge Institute for Research, Madison, WI 53715, USA. [2]Department of Biochemistry and Molecular Biophysics, Washington University School of Medicine, St. Louis, MO 63110, USA. [3]Department of Biomolecular Chemistry, University of Wisconsin-Madison, Madison, WI 53706, USA. [4]Department of Chemistry, University of Wisconsin-Madison, Madison, WI 53706, USA. [5]Department of Cell Biology and Physiology, Washington University School of Medicine, St. Louis, MO 63110, USA. [6]School of Biomedical Sciences, Faculty of Medicine, University of Queensland, Brisbane, QLD 4072, Australia. [7]Department of Cell Biology, University of Texas Southwestern Medical Center, Dallas, TX 75390, USA. [8]Department of Biochemistry, University of Wisconsin-Madison, Madison, WI 53706, USA. [9]National Center for Quantitative Biology of Complex Systems, Madison, WI 53706, USA. [10]The University of Queensland, Institute for Molecular Bioscience, Brisbane, QLD 4072, Australia. [11]The University of Queensland Diamantina Institute, Faculty of Medicine, The University of Queensland, Brisbane, QLD 4102, Australia. [12]Department of Genetics, Washington University School of Medicine, St. Louis, MO 63110, USA. ✉e-mail: niemi@wustl.edu; pagliarini@wustl.edu

