## [Peer Review File · Nature Communications]

PPTC7 maintains mitochondrial protein content by suppressing receptor-mediated mitophagyREVIEWER COMMENTS

Reviewer #1 (Remarks to the Author):

In the manuscript, the authors are presenting additional data on the role of phosphatase PPTC7 using a mouse model. Using two models (conditional KO and MEFs derived from classical KO) they have demonstrated that mitochondrial protein mass is dependent on receptor-mediated mitophagy. Although the topic is very interesting there are some concerns regarding the study:

- In the Fig 1D plot the authors only name BNIP3 and NIX as upregulated proteins in Pptc7 KO but it is clearly seen that other proteins show even more distinct differences in expression. What are other proteins? Are they related to mitophagy?
- To analyze mitophagy the authors only used mt-mKeima to monitor mitophagy progression using fluorescence microscopy. The images are of poor quality and merge images are not showing single stainings. More precise analysis with mt-mKeima is achieved with flow cytometry (check doi.org/10.1002/cpcb.99). In triple knockout there is reduced mitophagy compared to Pptc^{-/-} but more than wt. It is essential that other mitophagy/autophagy markers are tested. LC3 blots and evaluation of LC3-I and LC3-II ratio is needed – to show that mitophagy is indeed elevated. Western blots are preferred for this and LC3 staining should be accompanied by a mitochondrial protein marker to clearly show the mitochondrial mass in this experimental setup. Why single BNIP3 and NIX knockouts are not shown?
- In addition to the previous comment, rescue experiments are needed to show that indeed BNIP3 and/or NIX are responsible for global mitochondrial reduction due to increased mitophagy.
- What is going on with other mitophagy receptors? What is going on with Pink1/Parkin-dependent mitophagy? This should be addressed.
- The authors mention the global increase in phosphoproteins – this is expected when phosphatase is knocked-out but no functional data have been shown beyond the proteins they have already pointed out in their previous paper (Niemi et al 2019). BNIP3 and NIX are the most studied mitophagy receptors and a lot is already known about their regulation especially concerning their phosphorylation status (many papers published recently). The authors in this manuscript have not shown any new data – they only claim “These phosphorylation events span eight unique residues in each of the Bnip3 and Nix proteins, suggesting complex phosphorylation-based regulation” which is not novel. They do not name the sites at all. Also, BNIP3 and NIX are different enough in their amino acid sequence that their phosphosites should be studied separately. Furthermore, the sentence: “Nonetheless, the consistent upregulation of these phosphopeptides in Pptc7 KO systems suggests that Pptc7 influences this phosphorylation status, possibly through direct interaction.” – this should be shown in this manuscript if to be accepted to this journal. Moreover, the authors discuss this in details in the Discussion section “Bnip3 and Nix are also phosphorylated in Pptc7 knockout systems, consistent with a model in which phosphorylation promotes Bnip3- and Nix-mediated mitophagic signaling...”, but this has not been shown in this manuscript. More data is needed.

Minor:

- In lines 118-120, the authors draw too strong conclusions.
- It would be nicer if the figure panels are mentioned in the text in the same order as presented in the figures.
- Figure 3A is not mentioned in the text.

Reviewer #2 (Remarks to the Author):

The manuscript by Niemi et al described the changes in mitochondrial proteins and function by Pptc7 knockout in mice and MEF cells. This is based on the group's previous publication showing that Pptc7 mice die soon after birth. They have generated flox mice so that Pptc7 deletion could be achieved by induction. The major findings include Pptc7 deletion leads to more mitophagy and reduced mitochondrial function. Its role in mitophagy is likely mediated by two proteins Bnip3 and Nix, which are upregulated in the KO cells. Overall, the paper might be of interest to the audience in the field. The significance of Pptc7 gene in the adult is also unclear, since the phenotype of the inducible Pptc7 KO mice was not described. Many of the data are still preliminary, making it hard to draw solid conclusions. Specific questions are as follows:

- (1) Do inducible Pptc7 KO mice have any phenotypes?
- (2) Can mitophagy be confirmed by more markers besides mitokiemia imaging? Does Bnip3 or Bnip3l overexpression induce mitophagy?
- (3) Bnip3 and Nix phosphorylation was also shown to regulate cell death, any phenotypes in cell death in Pptc7 KO mice or cells?
- (4) The mitochondrial function study in Fig. 2 could be studied in isolated mitochondria. Otherwise, its functional decline in cells and reduced mitochondrial protein expression could be caused by reduced number of mitochondria.
- (5) Fig. 3D-E could be explained in more detail. What data showed that the mitochondrial proteins are substantially rescued in the TKO cells compared to the Pptc7 KO cells?
- (6) The mechanistic insight is still limited. Could Pptc7 regulate Bnip3 and Nix indirectly? Or could the authors provide more direct biochemical evidence that Pptc7 dephosphorylate these two proteins?
- (7) The upregulation of 4 phosphoisoforms in Pptc7 KO cells and tissues is interesting. Unfortunately, there is no further study on these 4 proteins in the manuscript.

Reviewer #3 (Remarks to the Author):

In the manuscript, "Pptc7 maintains mitochondrial protein content by suppressing receptor-mediated mitophagy" Niemi et al. show that loss of Pptc7, a resident mitochondrial phosphatase, leads to dysregulation of several metabolic pathways and a significant increase in mitophagy, resulting in a marked reduction in mitochondrial mass. Phosphoproteomics analyses in perinatal tissues, adult liver, and MEFs demonstrate a common set of elevated phosphosites, including multiple sites on Bnip3 and Nix. Bnip3 and Nix are mitophagy receptors which may explain the increased mitophagy in the absence of Pptc7. These data suggest that Pptc7 deletion causes mitochondrial dysfunction via dysregulation of several metabolic pathways and that Pptc7 may directly regulate mitophagy receptor function or stability. Overall, the paper highlights the importance of managing mitochondrial protein phosphorylation to maintain a healthy mitochondrial population. This is an important paper that has substantial physiological relevance.

Addressing the following questions/suggestions will significantly improve the manuscript:

- In the previous publication (Neimi et al. 2019), the group described the Pptc7 KO mouse with a severe phenotype and died 1 day after birth. That was the primary motivation for making the inducible line in this study. To establish the model, authors should add metabolic characterization of the inducible line. They should address viability, growth, and metabolic status (does the mature KO suffer from hypoketotic hypoglycemia? Or was that a developmental issue). Also, the authors compare the proteome of the two KO models (Fig 1E) and claim a correlation, although R2 demonstrates a low correlation. Could the proteins that are not shared be linked to developmental pathways?
- Figure 1D: What proportion of mitochondrial proteins were up or downregulated in the KO? How many proteins were identified (whole proteome, mitochondria)? And out of those, how many are up and down? Please discuss changes to the non-mitochondrial proteome, if any.
- It was recently proposed in yeasts that the accumulation of phosphorylated proteins, rather than the phosphorylation of a specific protein, signal for mitophagy (Kolitsida, P., Zhou, J., Rackiewicz, M., Nolić, V., Dengjel, J. and Abeliovich, H. (2019). Phosphorylation of mitochondrial matrix proteins regulates their selective mitophagic degradation. Proc. Natl. Acad. Sci. U.S.A). Did you evaluate the overall phosphoproteome in the Pptc7 KO?
- Figure 3B: Can you show what FCCP mitophagy looks like with this assay? Also, can you visualize the reduced levels of mitochondria with MitoTracker Green? The images give the impression that the mitochondria look comparable in the 458 channel.
- Providing EM images of mitochondria engulfed in a degradative compartment in these cells will be beneficial. Also, is mitochondrial morphology altered in Pptc7 KO and rescued by Bnip3/Nix depletion?
- Figure 3D and E: the rescue is not clear in the volcano plot, maybe color coding can help?
- Figure 4A and B: what happens to the whole cell phosphoproteome?
- Is general autophagy affected in the Pptc7 cells and is there an increase in cellular stress markers?

- Pptc7 mitochondrial phosphatase, Binp3, and Nix are on the outer mitochondrial membrane. Please discuss potential mechanisms/ideas for physical interactions or Pptc7 impact on their stability.
- What is the potential alternative mechanism by which Pptc7 may regulate mitochondrial function?

Reviewer #4 (Remarks to the Author):

The work demonstrates the role of the phosphatase Pptc7 in regulating mitochondrial protein content by altering phosphorylation dynamics. The authors also point out the mitophagy receptors Binp3 and Nix as Pptc7 substrates that undergo direct dephosphorylation. The hypothesis and experiments to test various hypothesis are well laid out. I have a few questions about the phosphoproteomic analysis.

-What was the specificity of the phosphoenrichment? Or what % of the peptide pool were phosphopeptides?

- Line 483: Phosphorylation sites were considered localized if they delivered MaxQuant localization scores >0.75. Is this the cut-off for including or excluding a phosphosite identification? If so, what's the reason for selecting this cut off and do the phosphosites identified (overlapping and unique) get altered if it is set higher, say 0.90?

- How were the figures generated? Which software was used for making plots? Including those details would make the Methods section more complete.

Response to reviewers

We thank all four reviewers for their insightful comments and lines of questioning, which we feel have substantially elevated our manuscript and its findings. In response to reviewer requests, the following figure panels have been added to the revised manuscript:

Figure 1F – Gene function analysis of upregulated non-mitochondrial proteins in *Pptc7* knockout liver tissue

Figure 1G – Quantification of serum ketones in fasted *Pptc7* knockout male mice

Figure 1H – Quantification of liver triacylglycerols in *Pptc7* knockout male and female liver

Figure 1I – Analysis of perilipin protein expression (PLIN2-PLIN5) in *Pptc7* knockout male and female liver

Figure 2H – Seahorse analysis of palmitate-BSA response in wild-type and *Pptc7* knockout MEFs

Figure 3D – Quantification of mt-Keima via FACS analysis in wild-type, *Pptc7* KO, and TKO MEFs

Figure 3E – Alternative presentation of mitochondrial proteome rescue in TKO cells; additional data now include analysis of single *Bnip3* and *Bnip3l* knockout cells

Figure 4F – Schematic of identified phosphorylation sites on BNIP3 and NIX

Figure 4G – Table including all identified phosphoresidues on BNIP3 and NIX as well as tissue-specific IDs

Figure 5A – Non-self-interactome of PPTC7 via affinity purification-mass spectrometry

Figure 5B – Immunoprecipitations of overexpressed PPTC7 with endogenous BNIP3 and NIX

Figure 5C – Yeast 2 hybrid analysis of PPTC7, BNIP3, and NIX reciprocal interactions

Figure 5D – Phosphoproteomic analysis of BNIP3 and NIX on mitochondria isolated from *Pptc7* knockout MEFs treated with recombinant PPTC7 or a catalytically inactive mutant (PPTC7 D78A)

Supplemental Figure 1C – Analysis of upregulated MitoCarta proteins in response to *Pptc7* knockout

Supplemental Figure 1G – Quantification of hepatic cholesterol levels in *Pptc7* knockout mice

Supplemental Figure 1H – Quantification of serum triacylglycerol levels in *Pptc7* knockout mice

Supplemental Figure 1I – Quantification of serum cholesterol levels in *Pptc7* knockout mice

Supplemental Figure 3A – FACS histograms of mt-Keima in wild-type, *Pptc7* KO, and TKO MEFs

Supplemental Figure 3B – Quantification of mt-Keima in wild-type MEFs overexpressing BNIP3 and mutants

Supplemental Figure 3C – Quantification of mt-Keima in wild-type MEFs overexpressing NIX

Supplemental Figure 3D – Analysis of proteomic data for receptor-mediated mitophagy proteins, ubiquitination-mediated mitophagy proteins, and general autophagy proteins from *Pptc7* KO liver (top panels) and *Pptc7* KO MEFs (bottom panels).

Supplemental Figure 3E – Table reporting the fold changes and p-values of all quantified proteins involved in receptor-mediated mitophagy, ubiquitination-mediated mitophagy, and general autophagy.

Supplemental Figure 3F – Western blot showing kinetic analysis of LC3-I and LC3-II accumulation in response to bafilomycin A in wild-type and *Pptc7* KO MEFs.

Supplemental Figure 4A – Analysis of phosphoproteomic data of non-mitochondrial phosphoisoforms in inducible *Pptc7* KO liver tissue

Supplemental Figure 4B – Analysis of phosphoproteomic data of non-mitochondrial phosphoisoforms in *Pptc7* KO MEFs

Additionally, we have provided the following figures to address reviewer concerns within the rebuttal:

Rebuttal Figure 1 – Quantified apoptotic proteins in *Pptc7* knockout liver and MEFs

Rebuttal Figure 2 – EM of liver tissue from *Pptc7* KO animals showing a mitochondrion engulfed in a double membraned structure (consistent with an autophagosome).

We are providing a point-by-point response to each of the reviewers' comments, shown in blue below. We also have used blue text within the manuscript to annotate differences between the original and revised manuscripts.

Reviewer #1

In the manuscript, the authors are presenting additional data on the role of phosphatase PPTC7 using a mouse model. Using two models (conditional KO and MEFs derived from classical KO) they have demonstrated that mitochondrial protein mass is dependent on receptor-mediated mitophagy. Although the topic is very interesting there are some concerns regarding the study:

- In the Fig 1D plot the authors only name BNIP3 and NIX as upregulated proteins in *Pptc7* KO but it is clearly seen that other proteins show even more distinct differences in expression. What are other proteins? Are they related to mitophagy?

This is an important observation, and we thank the reviewer for bringing this point to our attention. We reanalyzed our proteomics dataset reported in Figure 1D for proteins that are elevated and decreased in expression in both the mitochondrial and non-mitochondrial proteome. We have added details on this analysis to our manuscript, which can be found in the second paragraph under the header “PPTC7 maintains mitochondrial protein levels in adult mice”. This text reads as follows:

“In this experiment, we identified 6,749 proteins with 1,367 proteins demonstrating significant alterations in *Pptc7*^{-/-} liver relative to control tissue. Approximately equal numbers of proteins significantly increased (646) or decreased (721) in abundance. Notably, when stratified for mitochondrial localization, only 14 proteins in Mitocarta 3.0 increased over a log₂ fold change of 0.2 in *Pptc7* KO liver, while 512 mitochondrial proteins decreased beyond a log₂ fold change of -0.2 (Supplemental Figure 1C).”

As requested, we have listed the 14 significantly elevated mitochondrial proteins and their functions, as annotated in Mitocarta 3.0, in Supplemental Figure 1C. Importantly, to our knowledge, none of the significantly elevated mitochondrial proteins are related to mitophagy, consistent with our hypothesis that BNIP3 and NIX drive the elevated mitophagy seen in *Pptc7* knockout systems. However, we recognize that it is possible that non-mitochondrial proteins could influence mitophagy or autophagy in *Pptc7* knockout tissue. We thus also analyzed the fold changes of other proteins known to be involved in mitophagy and autophagy in our liver and MEF proteomics datasets. We quantified multiple other proteins in both the receptor-mediated and ubiquitination-linked mitophagy pathways but did not find any of these proteins to be significantly elevated (Supplemental Figures 3G-H). Furthermore, we identified fifteen proteins associated with autophagy and do not find significant deviations in these proteins in *Pptc7* KO liver tissue relative to wild-type liver tissue (Supplemental Figure 3D-E). These data demonstrate that there is no significant elevation in quantified autophagy and mitophagy proteins beyond *Bnip3* and *Nix* in either dataset, at least at basal levels.

We have additionally performed analysis on elevated non-mitochondrial proteins in response to this line of questioning, which was also brought up by Reviewer #3. We have performed gene ontology analysis on non-mitochondrial proteins identified as significantly elevated in *Pptc7* knockout liver tissue. The results from this analysis demonstrated an enrichment in proteins involved in fatty acid oxidation and lipid homeostasis. As our previously study of global *Pptc7* knockout mice also suggested defects in fatty acid homeostasis, we tested whether *Pptc7* knockout liver tissue manifested metabolic defects that could be associated with lipid oxidation or storage. We find that *Pptc7* KO mice have elevated hepatic triacylglycerols (TAGs), consistent with steatosis, but do not have an altered ketogenic response to an overnight fast (Figure 1G), nor do they show significant differences in serum TAGs, serum cholesterol, or hepatic cholesterol (Supplemental Figures 1G-I).

- To analyze mitophagy the authors only used mt-mKeima to monitor mitophagy progression using fluorescence microscopy. The images are of poor quality and merge images are not showing single stainings. More precise analysis with mt-mKeima is achieved with flow cytometry (check doi.org/10.1002/cpcb.99). In triple knockout there is reduced mitophagy compared to *Pptc7*^{-/-} but more than wt. It is essential that other mitophagy/autophagy markers are tested. LC3 blots and evaluation of LC3-I and LC3-II ratio is needed – to show that mitophagy is indeed elevated. Western blots are preferred for this and LC3 staining should be accompanied by a mitochondrial protein marker to clearly show the mitochondrial mass in this experimental setup. Why single BNIP3 and NIX knockouts are not shown?

We appreciate the reviewer encouraging a more thorough analysis of the mitophagy phenotype examined in this manuscript, which was also requested by Reviewer #2. However, we respectfully disagree that the images of mt-Keima shown in Figure 3B are of poor quality and note that images of single staining of mt-Keima at both physiological and acidic pH are included in the top and middle row of the original figure, and these are the stains represented within the merge image within the bottom row.

As requested, we have performed flow cytometry analysis of wild-type, *Pptc7* KO, and both *Bnip3/Nix/Pptc7* TKO MEF cell lines. Similar to the microscopy results, we find a significant increase in the number of cells with acidic mt-Keima signal (i.e., 'high mitophagy' cells) in the *Pptc7* KO MEFs relative to wild-type cells, and this is rescued in both TKO clones (shown in Figure 3D and Supplemental Figure 3A). Importantly, the distribution of the *Pptc7* KO cells is similar in magnitude to wild-type fibroblasts overexpressing either BNIP3 or NIX – an experiment requested by Reviewer #3. These data are shown in Supplemental Figures 3B and 3C. These data are consistent with a model in which overexpression of *Bnip3* or *Nix* is sufficient to elevate mitophagy in MEFs, which is indeed what we see in *Pptc7* KO cells.

We have additionally performed experiments to determine the levels of LC3-I and LC3-II as requested. While *Pptc7* KO cells have elevated LC3-II levels that accumulate in response to bafilomycin A, as shown in Supplemental Figure 3F, we interpret this as an upregulation of general autophagic machinery, which likely works in concert with elevated BNIP3 and NIX to increase mitochondrial protein turnover. This data is consistent with previous studies that found that BNIP3 overexpression is sufficient to elevate autophagy (Bellot et al. Mol Cell Biol 2009 PMID: 19273585, Rikka et al., Cell Death Differ, 2011 PMID: 21278801). Furthermore, it suggests LC3 and BNIP3 facilitate mitophagy, consistent with a previous study demonstrating that BNIP3 interacts with LC3 but not GABARAP proteins to facilitate mitophagy (Hanna et al., J. Biol. Chem 2012 PMID: 22505714).

While we appreciate the request of the reviewer to include markers of mitochondrial mass to accompany the LC3-II western blots, we note that decreases in mitochondrial proteins are not easily visualized by western blot in our *Pptc7* knockout cell culture systems. As shown in the MEF proteomics experiment (Figure 2), the fold change in the mitochondrial proteome is small compared to that in overall mouse tissues, with the 'average' downregulated mitochondrial protein manifesting a \log_2 fold change of -0.103 (range: -0.63 to -0.00036 amongst downregulated mitochondrial proteins). We suggest that proteomic evaluation of mitochondrial protein content is more robust than evaluation of individual mitochondrial proteins by western blot due to power in numbers; though each individual fold change is relatively small, the trend across most mitochondrial proteins is consistent with diminished mitochondrial protein content. This is further bolstered by the reproducibility of this phenotype in our study of the liver tissue of adult mice, as shown in this manuscript, as well as in perinatal heart and liver tissue, as shown in our previous study (Niemi et al. *Nat Commun*, 2019).

To address the question regarding why single *Bnip3* and *Nix* knockout cell lines were not evaluated, we have generated these knockout cells and have performed proteomic analyses on *Bnip3/Pptc7* and *Nix/Pptc7* knockout cell lines as requested. These data demonstrate that these knockout lines carrying knockouts of *Bnip3* or *Nix* also rescue mitochondrial content in the *Pptc7* knockout background, with these data now shown in Figure 3D.

- In addition to the previous comment, rescue experiments are needed to show that indeed BNIP3 and/or NIX are responsible for global mitochondrial reduction due to increased mitophagy.

While rescuing the TKO cells with BNIP3 and NIX would demonstrate the specificity of the mitophagic response in *Pptc7* knockout cells, we feel as though the additional analysis requested by multiple reviewers and put forth in this rebuttal have addressed the specificity of the mitophagic response in *Pptc7* knockout cells. First, we have confirmed the elevation in mitophagy using FACS as requested by Reviewer #1. These data demonstrate an almost complete rescue of elevated mitophagy in *Pptc7/Bnip3/Bnip3l* TKO cells relative to *Pptc7* cells, showing by an orthogonal method that BNIP3 and NIX drive the elevated mitophagy seen in response to loss of *Pptc7* (Figure 3D and Supplemental Figure 3A). Second, in response to reviewer #2, we have shown that BNIP3 and NIX, when overexpressed, are sufficient to significantly elevate mitophagic flux as assayed by mt-Keima FACS (Supplemental Figures 3B and C). Third, we have extensively reanalyzed our proteomics data as requested by the reviewer and find that BNIP3 and NIX, but no other mitophagy or autophagy proteins, are consistently elevated across *Pptc7* knockout systems. Fourth, we previously queried our MITOMICS dataset to demonstrate that BNIP3 and NIX elevation are specific to *PPTC7* knockout cells amongst over 200 HAP1 CRISPR lines. Fifth,

we have found that BNIP3, and in some tissues or cell types, NIX, are consistently elevated across three different experimental models, four different cell and tissues types, and two distinct developmental states. Collectively, these data suggest that BNIP3 and NIX, and not other mitophagy proteins, drive the elevation in mitophagy seen in *Pptc7* knockout cells.

- What is going on with other mitophagy receptors? What is going on with Pink1/Parkin-dependent mitophagy? This should be addressed.

We have re-examined our proteomic datasets in both mouse liver and mouse embryonic fibroblasts to more carefully evaluate the mitophagy and general autophagy pathway responses to *Pptc7* knockout. This analysis is now included in Supplemental Figure 3D-E, with six volcano plots showing proteins identified in receptor-mediated mitophagy, ubiquitination-mediated mitophagy, and general autophagy (Supplemental Figure 3D) as well as a table reporting the names, fold changes, and p-values of all proteins identified in this analysis (Supplemental Figure 3E). In these experiments, we quantified seven proteins previously implicated in receptor-mediated mitophagy (BCL2L13, BNIP3, NIX, FKBP8, PGAM5, PHB, PHB2); six proteins implicated in ubiquitination-mediated mitophagy pathways (NBR1, OPTN, PARL, p62/SQSTM1, TAX1BP1, TBK1), and fifteen proteins implicated in general autophagy (ATG12, ATG16L1, ATG2A, ATG3, ATG4A, ATG4B, ATG5, ATG7, ATG9A, BECN1, GABARAP, GABARAPL1, GABARAPL2, and RHEB). Across all twenty-eight of these identified proteins, only BNIP3 and NIX are significantly elevated across both *Pptc7* knockout datasets (i.e., in both liver and MEFs). While some proteins, such as BCL2L13 and PHB, are significantly altered by loss of *Pptc7* in mouse liver, these proteins are *downregulated* relative to wild-type tissues, which is inconsistent with the model that they promote the elevated mitophagy phenotypes seen in *Pptc7* knockout cells. Some important mediators of mitophagy, such as NDP52, PINK1, and Parkin were not identified in our proteomics dataset. As it is well established that PINK1 is constitutively degraded in the absence of select mitochondrial stresses, these data may reflect this absence of stabilized PINK1. While a lack of a signal does not necessarily mean that a protein is not present, we suggest that the breadth of proteins quantified in this analysis across two independent experimental systems strongly support a model in which BNIP3 and NIX are responsible for the upregulation of mitophagy in response to *Pptc7* knockout. These data are further bolstered by our data in *Pptc7/Bnip3/Bnip3l* triple knockout cells, which largely rescue the elevated mitophagy seen in *Pptc7* knockout cells, our MITOMICs data, which demonstrate that BNIP3 and NIX are uniquely elevated in *Pptc7* knockout cells relative to over two hundred other HAP1 knockout cell lines, and our previous study (Niemi et al., Nat Commun, 2019), which identified BNIP3 as upregulated in the heart and liver tissues of mice harboring a global deletion of *Pptc7*. It is worth noting that we have found BNIP3 and/or NIX upregulation across three independent studies, three independent model systems, and five independent cell and/or tissue types. This reproducibility, along with the supportive data in this manuscript, strongly suggest that BNIP3 and NIX, and not other redundant pathways, drive mitophagy in *Pptc7* knockout cells.

- The authors mention the global increase in phosphoproteins – this is expected when phosphatase is knocked-out but no functional data have been shown beyond the proteins they have already pointed out in their previous paper (Niemi et al 2019). BNIP3 and NIX are the most studied mitophagy receptors and a lot is already known about their regulation especially concerning their phosphorylation status (many papers published recently). The authors in this manuscript have not shown any new data – they only claim “These phosphorylation events span eight unique residues in each of the Bnip3 and Nix proteins, suggesting complex phosphorylation-based regulation” which is not novel. They do not name the sites at all. Also, BNIP3 and NIX are different enough in their amino acid sequence that their phosphosites should be studied separately.

We realize that our initial presentation of BNIP3 and NIX phosphorylation events identified in our *Pptc7* knockout systems was unclear. We also acknowledge that a recent paper, which we cite within our manuscript, links two phosphorylation sites on BNIP3 to elevated stability. However, we identify eight phosphorylation sites on BNIP3 and eight phosphorylation sites on NIX – sixteen in total – the majority of which have not been studied in detail to our knowledge. We maintain that the fact that BNIP3 and NIX each can be phosphorylated across eight independent residues, each of which have uniquely predicted kinase consensus motifs, does indeed suggest “complex, phosphorylation-based regulation.”

We apologize for not naming these sixteen phosphorylation sites in our initial draft and appreciate that this may not have fully communicated such complexities in our first draft. To address this, we have amended Figure 4 by

adding two subpanels. First, as noted by the reviewer, BNIP3 and NIX have distinct amino acid sequences and indeed their phosphorylation sites fall in different patterns across these proteins. We have thus shown a schematic of each of the sixteen identified phosphorylation sites and their relative locations across the primary sequences of BNIP3 and NIX in a cartoon schematic in Figure 4F. Additionally, we have added a table listing each of the phosphorylation sites identified across all four of our *Pptc7* knockout phosphoproteomic datasets, as well as the experimental condition in which they were identified as significantly elevated, in Figure 4G.

To our knowledge, this dataset represents the most complete identification of phosphorylation sites on BNIP3 and NIX. We agree with the reviewer that each of these phosphosites should be studied separately. However, because these phosphorylation sites occur in seemingly tissue-specific patterns, and we are unable to resolve whether combinatorial phosphorylation events occur on each of these proteins, we propose that a rigorous analysis of each of these phosphorylation sites (alone or in combination) is beyond the scope of the current manuscript. In lieu of initiating studies on each of these phosphorylation events in isolation, we set up an experiment to test whether PPTC7 could directly dephosphorylate BNIP3 and NIX, which would additionally allow us to test if the phosphatase possessed specificity toward any of the identified phosphorylation events. We found that recombinant PPTC7 dephosphorylated most identified BNIP3 and NIX phosphorylation sites identified in *Pptc7* KO MEF mitochondria (Figure 5D). These studies thus motivate a more complete and thorough investigation of each of these phosphorylation events and their downstream effects on mitophagy, which we plan to perform in the future.

Furthermore, the sentence: “Nonetheless, the consistent upregulation of these phosphopeptides in *Pptc7* KO systems suggests that *Pptc7* influences this phosphorylation status, possibly through direct interaction.” – this should be shown in this manuscript if to be accepted to this journal.

We thank the reviewer for this suggestion and agree that this is a critical piece of data to support our proposed model. We have performed extensive experimentation to address this suggestion, adding a completely new figure (Figure 5) to address this point specifically. In Figure 5, we summarize data previously collected in the Pagliarini lab that showed that BNIP3 and NIX are the highest scoring interaction partners of PPTC7 in two independent cell types via affinity purification-mass spectrometry (Figure 5A). We confirm these interactions, both basally and in pseudohypoxic DFP-treated conditions, through immunoprecipitation of FLAG-tagged PPTC7 with endogenous BNIP3 and NIX (Figure 5B). As immunoprecipitations and affinity purification techniques do not demonstrate a direct interaction, we also cloned PPTC7, BNIP3, and NIX into yeast 2 hybrid vectors and showed strong, reciprocal interactions between these prey-bait pairs (Figure 5C). Finally, as mentioned in the point above, we demonstrated that recombinant PPTC7 can directly interact with and dephosphorylate BNIP3 and NIX on mitochondria isolated from *Pptc7* KO MEFs (Figure 5D). Collectively, these data demonstrate that PPTC7 has high affinity toward BNIP3 and NIX, and that this phosphatase can directly and functionally interact with these mitophagy receptors. These data do not demonstrate that such interactions occur within the cellular environment, a limitation that we discuss within both the results and the discussion of our manuscript. We are keen to follow up these studies and study the sublocalization of PPTC7 in greater detail in future studies.

Moreover, the authors discuss this in details in the Discussion section “*Bnip3* and *Nix* are also phosphorylated in *Pptc7* knockout systems, consistent with a model in which phosphorylation promotes *Bnip3*- and *Nix*-mediated mitophagic signaling....”, but this has not been shown in this manuscript. More data is needed.

We indeed state that our data are consistent with a model in which BNIP3 and NIX hyperphosphorylation promotes mitophagic signaling, and we believe that the data in this manuscript support such a model. First, we show that BNIP3 and NIX are hyperphosphorylated across four independent *Pptc7* knockout systems in Figure 4. In data now included within this revision, we further demonstrate that recombinant PPTC7 protein can dephosphorylate most of these phosphorylation sites, supporting a model in which the phosphatase can directly regulate these phosphosites. We additionally show that *Pptc7* knockout cells have elevated mitophagy relative to wild-type cells, and that this elevation in mitophagy is ablated through the knockout of BNIP3 and NIX (Figure 3). We have now shown this by microscopy and FACS analysis in two independent triple knockout clones (Figure 3). As requested by the reviewer, we have additionally reanalyzed our data, and the results for these analyses suggest that no other mitophagy or autophagy proteins are significantly upregulated in basal conditions in our proteomics data. However, we do show that one autophagic marker, LC3-II, accumulates to a larger extent in

Pptc7 knockout cells than in wild-type cells in response to bafilomycin A, suggesting that *Pptc7* knockout induces elevated autophagic flux. Collectively, we propose that these data do support a model in which phosphorylation promotes BNIP3- and NIX-mediated mitophagic signaling. However, we agree with the reviewer that further investigation of these phenotypes, particularly with respect to understanding the contributions of each independent phosphorylation site to BNIP3- and NIX-mediated function, are warranted, and we are eager to study this in the future.

Minor:

- In lines 118-120, the authors draw too strong conclusions.

We have amended this text to read: “Collectively, these data demonstrate that acute loss of *Pptc7* in adult mouse liver globally decreases mitochondrial content, suggesting that PPTC7 expression is required across *at least two physiological states* to maintain mitochondrial protein homeostasis.” Added text is italicized.

- It would be nicer if the figure panels are mentioned in the text in the same order as presented in the figures.

To our knowledge, the figure panels are mentioned in the text in the order presented in the figures, with the exception that we at times refer to previously discussed figures later within the text.

- Figure 3A is not mentioned in the text.

Thank you for catching this; this has been fixed.

Reviewer #2

The manuscript by Niemi et al described the changes in mitochondrial proteins and function by *Pptc7* knockout in mice and MEF cells. This is based on the group’s previous publication showing that *Pptc7* mice die soon after birth. They have generated flox mice so that *Pptc7* deletion could be achieved by induction. The major findings include *Pptc7* deletion leads to more mitophagy and reduced mitochondrial function. Its role in mitophagy is likely mediated by two proteins Bnip3 and Nix, which are upregulated in the KO cells. Overall, the paper might be of interest to the audience in the field. The significance of *Pptc7* gene in the adult is also unclear, since the phenotype of the inducible *Pptc7* KO mice was not described. Many of the data are still preliminary, making it hard to draw solid conclusions. Specific questions are as follows:

- (1) Do inducible *Pptc7* KO mice have any phenotypes?

We thank the reviewer for inquiring about the physiological responses to *Pptc7* loss in vivo. We note that we performed our proteomics analysis only two weeks post-tamoxifen-induced knockout, which may limit the severity of the phenotypes seen in the knockout mice. Despite this, our data suggest that substantial molecular changes occur in mouse liver at this timepoint, and a re-evaluation of the non-mitochondrial proteome as requested by Reviewer #3 suggested that proteins involved in fatty acid oxidation and general lipid handling are altered in response to loss of *Pptc7*. This was particularly interesting to us, as our previous study demonstrated that global loss of *Pptc7* causes perinatal lethality accompanied by hypoketotic hypoglycemia – a diagnostic hallmark of fatty acid oxidation disorders. We thus tested if the inducible *Pptc7* KO mice had compromised serum ketones upon an overnight fast, finding that male *Pptc7* KO mice do not have significantly altered serum ketones relative to wild-type mice (Figure 1G). As disrupted FAO has also been linked to elevated hepatic triacylglycerols (TAGs), we quantified TAGs from liver tissue isolated from control and *Pptc7* knockout animals and find a significant elevation of these lipids in the knockout animals of both sexes relative to their control littermates. We also tested a handful of other metabolic parameters, including serum TAGs, serum cholesterol, and hepatic cholesterol levels, but find no significant changes in any of these metabolites in *Pptc7* knockout animals relative to their control littermates. These data can be found in Supplemental Figure 1G-I.

- (2) Can mitophagy be confirmed by more markers besides mitokiemia imaging? Does Bnip3 or Bnip3l overexpression induce mitophagy?

We appreciate the reviewer encouraging a more thorough analysis of the mitophagy phenotype examined in this manuscript, which was also requested by Reviewer #1. We have added a more thorough characterization of mt-Keima positive cells by performing FACS analysis of wild-type, *Pptc7* knockout, and *Pptc7/Bnip3/Bnip3l* triple knockout (TKO) cells. Similar to the microscopy data, we find that *Pptc7* knockout cells have significantly elevated acidic mt-Keima signal, consistent with elevated mitophagic flux, which is rescued in both TKO cell lines. These data are included in Figure 3D (quantification) and Supplemental Figure 3A (FACS plots).

We thank the reviewer for the excellent inquiry as to whether BNIP3 or NIX overexpression induce mitophagy. Indeed, our model suggests that overexpression of these mitophagy receptors should be sufficient to induce mitophagy to a similar degree as seen in *Pptc7* knockout cells. To address this, we stably expressed BNIP3, NIX, or a vector-only control in wild-type MEFs expressing the mt-Keima reporter. These data from these experiments demonstrate that overexpression of BNIP3 or NIX increases the mt-Keima reporter signal in cells. Our data suggest that overexpression of BNIP3 induces a modest but statistically significant increase in the proportion of cells undergoing high mitophagy. We thus also generated two separate mutants, BNIP3 W13A, which disrupts the LC3-interaction region (LIR motif), as well as BNIP3 G173A, which disrupts dimerization. Each of these mutants have been previously shown to disable BNIP3 function, and indeed, we see do not see increases in the frequency of high mitophagy cells upon expression of these two mutants. These data support a model in which wild-type, but not mutant BNIP3 induces modest increases in the number of cells undergoing high mitophagy. We additionally overexpressed NIX in wild-type MEFs expressing the mt-Keima reporter and find that NIX overexpression increases the population of cells undergoing high mitophagy to a similar extent as seen in *Pptc7* knockout cells. Collectively, these data support a model in which BNIP3 and NIX overexpression can induce mitophagic flux. These data are shown in Supplemental Figures 3B-C.

(3) *Bnip3* and *Nix* phosphorylation was also shown to regulate cell death, any phenotypes in cell death in *Pptc7* KO mice or cells?

This is a very interesting question. BNIP3 was originally identified as an interactor of the adenovirus E1B 19kDa protein – a protein with functional similarities to the anti-apoptotic BCL-2 protein. As such, the overexpression of BNIP3 and/or NIX has been implicated in the induction of intrinsic apoptosis. Accordingly, one might hypothesize that *Pptc7* knockout tissues or cells undergo elevated apoptosis due to the overexpression of these receptors. Upon querying our proteomics data, we find some interesting changes, including significantly decreased cell death mediator BAK1 in both *Pptc7* knockout systems (Rebuttal Figure 1). However, BAK1 is known to localize to mitochondria, rendering it unclear if BAK1 is selectively reduced in *Pptc7* knockout systems, or if this is an indirect effect due to the decrease in mitochondrial protein content we have seen across cell and tissue types. Furthermore, a handful of other quantified apoptotic proteins are significantly changed in *Pptc7* knockout mouse liver, including pro-survival proteins BCL2X and MCL1 as well as caspases 6 and 9 (Rebuttal Figure 1). Despite their significance, the fold changes are small, and it is difficult to determine how cell death would be affected by the levels of these proteins alone, as both pro-survival and pro-apoptotic proteins are significantly altered. It is

worth noting that we find no visual evidence of elevated basal cell death when culturing *Pptc7* knockout cells – a phenotype that would be expected should hyperphosphorylation and/or overexpression of BNIP3 and NIX promote intrinsic apoptosis. This could additionally be reflected by small number of significantly changing cell death proteins in *Pptc7* KO MEFs relative to wild-type cells (Rebuttal Figure 1).

Rebuttal Figure 1: Analysis of cell death proteins in proteomics data from *Pptc7* knockout MEF (left) and liver (right) tissues. Volcano plots of all quantified proteins in MEFs (left) or liver (right) shown as their fold change between *Pptc7* KO and wild-type samples. Orange dots correspond to cell death proteins including BAK1, BAX, BCL2L1, BID, CASP12, CASP2, CASP3, CASP6, CASP7, CASP8, CASP9, MCL1, RIPK1, and RIPK3. Significantly altered proteins ($p < 0.05$, Student's T-test) are labeled.

Interestingly, some data suggest that BNIP3 and NIX overexpression may not always promote cell death, but may actually promote resistance to

some treatments that induce intrinsic apoptosis. A recent preprint used CRISPR screening to identify pathways that sensitize human fibroblasts to cell death triggered by the BH3-only mimetic ABT-263 (Colville et al., bioRxiv 2022, <https://doi.org/10.1101/2022.04.01.486768> doi). This screen identified two major hits – *PPTC7* and *FBXL4* – which, when knocked down, promote resistance to cell death induced by ABT-263. Notably, *FBXL4* was recently identified as an E3 ligase whose expression is critical for turning over BNIP3 and NIX (Nguyen-Dien and Kozul et al., EMBO 2023, PMID: 37161784). These data suggest that the directionality of BNIP3 and NIX overexpression in regards to cell death is likely more complex than originally anticipated, and that these experiments are beyond the scope of the current manuscript. We are, however, excited about exploring the role of *PPTC7* and BNIP3/NIX-mediated cell death phenotypes in the future.

(4) The mitochondrial function study in Fig. 2 could be studied in isolated mitochondria. Otherwise, its functional decline in cells and reduced mitochondrial protein expression could be caused by reduced number of mitochondria.

This is an excellent point, and one that we have thought about extensively. As the reviewer notes, there are multiple models consistent with our data demonstrating a decline in mitochondrial function in *Pptc7* knockout cells. One such model is that there are equivalent ‘numbers’ of mitochondria in wild-type and *Pptc7* knockout cells, but that the *Pptc7* KO mitochondria are less functional than wild-type mitochondria. An alternative model, as the reviewer notes, is that *Pptc7* knockout cells have fully functional mitochondria, but have fewer of them relative to wild-type cells. The experiment proposed by the reviewer should distinguish these models, as isolation of mitochondria allows for normalization of their content, thus eliminating this as a potential contributor to the dysfunctional mitochondrial phenotype seen in *Pptc7* knockout cells.

As such, we have optimized the isolation of mitochondria from mouse embryonic fibroblasts (MEFs) for this line of experimentation, which we also performed for the phosphoproteomics experiment now shown in Figure 5D. While we were able to enrich mitochondria using a highly cited and well-established protocol (Frezza et al. Nat Prot 2007, PMID: 17406588), we were unable to determine the relative fraction of mitochondria (versus other organellar contaminants, such as ER) in these experiments. In our experience in multiple rounds of mitochondrial isolation, we feel these ratios are variable and thus lead to inconsistent results. A better way to perform these experiments would be to further gradient purify these mitochondria for functional analysis. However, due to the extensive number of plates required for these experiments (we pooled 75x 15 cm² plates for a single phosphoproteomics experiment) and the need to optimize Percoll gradient purification as well as Seahorse analysis with isolated mitochondria (which requires optimization of mitochondrial seeding densities as well as reoptimization of compound dosages), we feel as though rigorously testing this question is beyond the scope of this manuscript. We do, however, agree with the reviewer that this is a critical point, and we plan to perform these experiments in both cells as well as tissue models in the future.

(5) Fig. 3D-E could be explained in more detail. What data showed that the mitochondrial proteins are substantially rescued in the TKO cells compared to the *Pptc7* KO cells?

We thank the reviewer for identifying this figure presentation as problematic—a point that was also raised by Reviewer #3. We previously plotted the rescue data as a volcano plot, which was not easily interpretable by readers due to the *Pptc7* KO/WT data being presented in a separate figure. We have replaced these volcano plots with dot plots showing the fold change of all identified mitochondrial proteins in *Pptc7* KO/WT conditions as well as rescue comparisons of each single knockout (i.e., *Bnip3* KO/*Pptc7* KO and *Bnip3l* KO/*Pptc7* KO) as well as both triple knockout, or TKO, clones (i.e., *Bnip3/Bnip3l/Pptc7* KO #1/*Pptc7* KO and *Bnip3/Bnip3l/Pptc7* KO #2/*Pptc7* KO). Each of these latter four comparisons shows a significant increase in mitochondrial proteins relative to the *Pptc7* KO/WT condition, suggesting a significant rescue of the mitochondrial proteome across conditions relative to the *Pptc7* KO/WT comparison. This new presentation of data is shown in Figure 3D.

(6) The mechanistic insight is still limited. Could *Pptc7* regulate *Bnip3* and *Nix* indirectly? Or could the authors provide more direct biochemical evidence that *Pptc7* dephosphorylate these two proteins?

In this revision, we have provided substantial additional experimentation to explore the mechanistic interactions between *PPTC7* and BNIP3 and NIX. Evidence that *PPTC7* could directly interact with BNIP3 and NIX was also requested by Reviewer #1, and we have included four new datasets in Figure 5 to address this point specifically.

In Figure 5, we summarize data previously collected in the Pagliarini lab that showed that BNIP3 and NIX are the highest scoring interaction partners of PPTC7 in two independent cell types via affinity purification-mass spectrometry (Figure 5A). We confirm these interactions, both basally and in pseudohypoxic DFP-treated conditions, through immunoprecipitation of FLAG-tagged PPTC7 with endogenous BNIP3 and NIX (Figure 5B). As immunoprecipitations and affinity purification techniques do not demonstrate a direct interaction, we also cloned PPTC7, BNIP3, and NIX into yeast 2 hybrid vectors and showed strong, reciprocal interactions between these prey-bait pairs (Figure 5C). Finally, as mentioned in the point above, we demonstrated that recombinant PPTC7 can directly interact with and dephosphorylate BNIP3 and NIX on mitochondria isolated from *Pptc7* KO MEFs (Figure 5D). Collectively, these data demonstrate that PPTC7 has high affinity toward BNIP3 and NIX, and that this phosphatase can directly and functionally interact with these mitophagy receptors. These data do not demonstrate that such interactions occur within the cellular environment, a limitation that we discuss within both the results and the discussion of our manuscript. We are keen to follow up these studies and study the sublocalization of PPTC7 in greater detail in future studies.

(7) The upregulation of 4 phosphoisoforms in *Pptc7* KO cells and tissues is interesting. Unfortunately, there is no further study on these 4 proteins in the manuscript.

We agree with the reviewer that these 4 phosphoisoforms that reproducibly increase in *Pptc7* KO cells are interesting and worth investigation. However, we have now included data on the interactions between PPTC7, BNIP3 and NIX, including data demonstrating that recombinant PPTC7 can directly dephosphorylate these mitophagy receptors. As these data suggest that BNIP3 and NIX themselves may be phosphosubstrates of PPTC7, we feel that extensive mechanistic investigation of these four phosphoisoforms is beyond the scope of the current manuscript.

Reviewer #3

In the manuscript, “*Pptc7* maintains mitochondrial protein content by suppressing receptor-mediated mitophagy” Niemi et al. show that loss of *Pptc7*, a resident mitochondrial phosphatase, leads to dysregulation of several metabolic pathways and a significant increase in mitophagy, resulting in a marked reduction in mitochondrial mass. Phosphoproteomics analyses in perinatal tissues, adult liver, and MEFs demonstrate a common set of elevated phosphosites, including multiple sites on *Bnip3* and *Nix*. *Bnip3* and *Nix* are mitophagy receptors which may explain the increased mitophagy in the absence of *Pptc7*. These data suggest that *Pptc7* deletion causes mitochondrial dysfunction via dysregulation of several metabolic pathways and that *Pptc7* may directly regulate mitophagy receptor function or stability. Overall, the paper highlights the importance of managing mitochondrial protein phosphorylation to maintain a healthy mitochondrial population. This is an important paper that has substantial physiological relevance.

We thank the reviewer for their positive feedback and comments on our work.

Addressing the following questions/suggestions will significantly improve the manuscript:

- In the previous publication (Niemi et al. 2019), the group described the *Pptc7* KO mouse with a severe phenotype and died 1 day after birth. That was the primary motivation for making the inducible line in this study. To establish the model, authors should add metabolic characterization of the inducible line. They should address viability, growth, and metabolic status (does the mature KO suffer from hypoketotic hypoglycemia? Or was that a developmental issue). Also, the authors compare the proteome of the two KO models (Fig 1E) and claim a correlation, although R2 demonstrates a low correlation. Could the proteins that are not shared be linked to developmental pathways?

We are interested in the metabolic effects of loss of *Pptc7* and were surprised to find the loss of this protein in adult mice resulted in no overt phenotypes in the first two weeks of gene loss (i.e., the time at which our phosphoproteomics study was performed). We have addressed viability in the manuscript by noting the differences between the global knockout model (which is perinatal lethal) and the inducible model (which survives up to two weeks post tamoxifen-induced loss of *Pptc7*). We did not track growth of the mice during the course of our experiment due to the short duration, as the mice were only housed for two weeks after *Pptc7* knockout. It is worth noting that tamoxifen treatment induces weight loss in mice, and the acute nature of our study would render it difficult to determine gene-specific growth phenotypes versus those that derive from the tamoxifen

treatment. In the future, we intend to do a more extensive metabolic and growth study on the adult inducible mice at timepoints beyond two weeks.

We appreciate the inquiry into the metabolic status of the *Pptc7* knockout animals and have addressed this in Figure 1 of our revised manuscript. More details on these studies can be found in the subsequent point, as these findings are tied to our analysis of the non-mitochondrial proteome as requested by this reviewer.

- Figure 1D: What proportion of mitochondrial proteins were up or downregulated in the KO? How many proteins were identified (whole proteome, mitochondria)? And out of those, how many are up and down? Please discuss changes to the non-mitochondrial proteome, if any.

We thank the reviewer for allowing us to clarify the analysis of our proteomics data, which was also requested by Reviewer #1. We reanalyzed our proteomics dataset reported in Figure 1D for proteins that are elevated and decreased in expression in both the mitochondrial and non-mitochondrial proteome. We have added details on this analysis to our manuscript, which can be found in the second paragraph under the header “PPTC7 maintains mitochondrial protein levels in adult mice”. This text reads as follows:

“In this experiment, we identified 6,749 proteins with 1,367 proteins demonstrating significant alterations in *Pptc7*^{-/-} liver relative to control tissue. Approximately equal numbers of proteins significantly increased (646) or decreased (721) in abundance. Notably, when stratified for mitochondrial localization, only 14 proteins in Mitocarta 3.0 increased over a log₂ fold change of 0.2 in *Pptc7* KO liver, while 512 mitochondrial proteins decreased beyond a log₂ fold change of -0.2 (Supplemental Figure 1C).”

We have additionally performed analysis on elevated non-mitochondrial proteins in response to this line of questioning, which has revealed interesting physiological insights that we explore in new experimentation now shown in Figure 1. We performed gene ontology analysis on non-mitochondrial proteins identified as significantly elevated in *Pptc7* knockout liver tissue. The results from this analysis demonstrated an enrichment in proteins involved in fatty acid oxidation and lipid homeostasis, which is now shown in Figure 1F. As our previously study of global *Pptc7* knockout mice also suggested defects in fatty acid homeostasis, we tested whether *Pptc7* knockout liver tissue manifested metabolic defects that could be associated with lipid oxidation or storage. We find that *Pptc7* KO mice have elevated hepatic triacylglycerols (TAGs), consistent with steatosis, but do not have an altered ketogenic response to an overnight fast (Figure 1G), nor do they show significant differences in serum TAGs, serum cholesterol, or hepatic cholesterol (Supplemental Figures 1G-I).

- It was recently proposed in yeasts that the accumulation of phosphorylated proteins, rather than the phosphorylation of a specific protein, signal for mitophagy (Kolitsida, P., Zhou, J., Rackiewicz, M., Nolić, V., Dengjel, J. and Abeliovich, H. (2019). Phosphorylation of mitochondrial matrix proteins regulates their selective mitophagic degradation. Proc. Natl. Acad. Sci. U.S.A). Did you evaluate the overall phosphoproteome in the *Pptc7* KO?

While we only reported the mitochondrial phosphoproteome in our *Pptc7* KO systems in Figure 4, we did simultaneously collect data on the whole cell phosphoproteome. While these data were included in our original supplemental dataset, we have also shown these data as Supplemental Figures 4A and 4B. While there are some interesting phosphorylation sites that have been identified in this analysis, most of them remain uncharacterized in terms of their molecular function. Thus, we have not commented upon any of these phosphorylation events within our manuscript, although some may be worthy of molecular follow up in the future. We have also added citations from the Abeliovich lab regarding mitophagic selectivity into the discussion section of our manuscript.

- Figure 3B: Can you show what FCCP mitophagy looks like with this assay? Also, can you visualize the reduced levels of mitochondria with MitoTracker Green? The images give the impression that the mitochondria look comparable in the 458 channel.

These are interesting questions and suggestions. As the reviewer is likely aware, FCCP is a well-known trigger for PINK1/parkin-mediated mitophagy. As we have performed more thorough analysis of both proteomics datasets and find no evidence of PINK1/parkin-mediated mitophagy activation in our cells, we are unclear what

the expected result from such an experiment may yield. Given our collected data on elevated LC3 flux (Supplementary Figure 3F), it is possible that FCCP treatment may indeed increase mitophagic flux in *Pptc7* knockout cells relative to wild-type cells, as autophagic flux is elevated. However, given that this may upregulate a parallel pathway for mitophagic induction (i.e., one that is independent of BNIP3 and NIX), it is also possible that there will be no significant difference in FCCP-induced mitophagic flux in wild-type and *Pptc7* knockout cells. In lieu of these experiments, we have looked at the contributions of directly overexpressing BNIP3 and NIX in wild-type cells to understand if these receptors are sufficient to elevate mitophagy, as requested by Reviewer #2. To address this, we stably expressed BNIP3, NIX, or a vector-only control in wild-type MEFs expressing the mt-Keima reporter. These data from these experiments demonstrate that overexpression of BNIP3 or NIX increases the mt-Keima reporter signal in cells. Our data suggest that overexpression of BNIP3 induces a modest but significant elevation in the proportion of cells undergoing high mitophagy. We thus also generated two separate mutants, BNIP3 W13A, which disrupts the LC3-interaction region (LIR motif), as well as BNIP3 G173A, which disrupts dimerization. Each of these mutants have been previously shown to disable BNIP3 function, and indeed, we see do not see increases in the number of cells undergoing high mitophagy upon expression of these two mutants. These data support a model in which wild-type, but not mutant BNIP3 induces modest elevation of cells undergoing high mitophagy. We additionally overexpressed NIX in wild-type MEFs expressing the mt-Keima reporter and find that NIX overexpression increases the population of cells undergoing high mitophagy to a similar extent as seen in *Pptc7* knockout cells. Collectively, these data support a model in which BNIP3 and NIX overexpression can induce mitophagic flux. These data are shown in Supplemental Figures 3B-C.

The experimental suggestion to visualize mitochondrial content by MitoTracker staining is an interesting one, and this is a point that we have thought about before. We agree that the mitochondrial levels in the 458 channel of our data presented in Figure 3 look similar, and we have previously undertaken MitoTracker staining to determine mitochondrial content and have not seen significant differences in these measurements. It is unclear to us whether this is technical (i.e., MitoTracker does not have sufficient resolution to reveal the differences in content seen in *Pptc7* knockout cells) or biological (i.e., mitochondrial 'content' per se is unchanged, but the levels of mitochondrial proteins themselves are decreased). As noted for Reviewer #1, the *Pptc7* knockout MEFs do not show large decreases in individual levels of mitochondrial proteins, and mitochondrial proteins are more robustly lost in *Pptc7* knockout tissues in vivo. We are interested in exploring this question further in *Pptc7* knockout tissues, likely by comparing wet weights of purified mitochondria between wild-type and knockout animals.

- Providing EM images of mitochondria engulfed in a degradative compartment in these cells will be beneficial. Also, is mitochondrial morphology altered in *Pptc7* KO and rescued by *BNIP3/Nix* depletion?

Rebuttal Figure 2: EM of *Pptc7* KO tissue. Tissue slices from *Pptc7* KO liver show mitochondria surrounded by a double membrane (white arrows) which may be an autophagosome.

We thank the reviewer for this suggestion. We agree the EM images of mitochondria undergoing mitophagy would make a powerful addition to our manuscript. To address this, we performed EM of liver sections from control (i.e., floxed) and *Pptc7* knockout mice. In *Pptc7* knockout liver, we indeed see mitochondria that are proximal to or engulfed in a double membraned compartment, which would be consistent with an autophagosome (Rebuttal Figure 2). However, we are not certain that these membranes are *bona fide* autophagosomes (i.e., they could derive from other organelle membranes) nor are we confident in their frequency in our samples. Thus, we have chosen not to report these data within our manuscript, but plan to explore this phenotype more thoroughly in future studies.

- Figure 3D and E: the rescue is not clear in the volcano plot, maybe color coding can help?

We thank the reviewer for this comment, as it was also brought up by Reviewer #2. We have changed the presentation of these data to a dot plot figure similar to that shown in Figure 2D. This new presentation, along with newly requested data on *Bnip3*^{-/-} and *Bnip3*^{+/+} single knockout lines, are displayed in Figure 3E.

- Figure 4A and B: what happens to the whole cell phosphoproteome?

As commented above, we have analyzed the non-mitochondrial phosphoproteome and show the trends in these phosphorylation events in Supplemental Figures 4A and 4B for mouse liver tissue and mouse embryonic fibroblasts as requested.

- Is general autophagy affected in the *Pptc7* cells and is there an increase in cellular stress markers?

We thank the reviewer for encouraging a more thorough analysis of autophagic markers, which was also requested by Reviewer #1. We have re-examined our proteomic datasets in both mouse liver and mouse embryonic fibroblasts to more carefully evaluate the mitophagy and general autophagy pathway responses to *Pptc7* knockout. This analysis is now included in Supplemental Figure 3D-E, with six volcano plots showing proteins identified in receptor-mediated mitophagy, ubiquitination-mediated mitophagy, and general autophagy (Supplemental Figure 3D) as well as a table reporting the names, fold changes, and p-values of all proteins identified in this analysis (Supplemental Figure 3E). In these experiments, we quantified seven proteins previously implicated in receptor-mediated mitophagy (BCL2L13, BNIP3, NIX, FKBP8, PGAM5, PHB, PHB2); six proteins implicated in ubiquitination-mediated mitophagy pathways (NBR1, OPTN, PARL, p62/SQSTM1, TAX1BP1, TBK1), and fifteen proteins implicated in general autophagy (ATG12, ATG16L1, ATG2A, ATG3, ATG4A, ATG4B, ATG5, ATG7, ATG9A, BECN1, GABARAP, GABARAPL1, GABARAPL2, and RHEB). Across all twenty-eight of these identified proteins, only BNIP3 and NIX are significantly elevated across both *Pptc7* knockout datasets (i.e., in both liver and MEFs).

Despite the fact that no general autophagic proteins are significantly upregulated across both proteomic datasets, we also analyzed LC3-II accumulation in the presence of bafilomycin A. These data demonstrate that LC3-II accumulates to a larger extent in *Pptc7* knockout cells than in wild-type cells in response to bafilomycin A, suggesting that *Pptc7* knockout induces elevated autophagic flux. These data are now reported in Supplemental Figure 3F.

- *Pptc7* mitochondrial phosphatase, *Bnip3*, and *Nix* are on the outer mitochondrial membrane. Please discuss potential mechanisms/ideas for physical interactions or *Pptc7* impact on their stability.

This is an interesting question, and one that we have begun to experimentally test with our new data as presented in Figure 5. We have not shown that *Pptc7* is on the outer mitochondrial membrane, instead referencing datasets that are suggestive of this possibility (such as the BioPlex datasets generated by Steve Gygi's group). While we believe that elucidating the mechanisms by which PPTC7 may be dual localized to be beyond the scope of this paper, we did feel it important to understand whether the interactions between BNIP3, NIX, and PPTC7 could happen, and if they were direct. To begin to understand this, we reanalyzed PPTC7 interactome data (Figure 5A), performed immunoprecipitations (Figure 5B), yeast 2 hybrid analysis (Figure 5C), and functional phosphoproteomics of *Pptc7* knockout mitochondria treated with recombinant PPTC7 (Figure 5D), which we hypothesize would mimic PPTC7 expressed on the outer mitochondrial membrane. Indeed, in each of these experiments, PPTC7 robustly associated with BNIP3 and NIX, suggesting these proteins can directly and functionally interact. We are keen to understand whether these interactions are maintained in the cellular context and under which conditions they might occur.

- What is the potential alternative mechanism by which *Pptc7* may regulate mitochondrial function?

This is an interesting question. As our phosphoproteomic analyses suggest that PPTC7 has substrates beyond BNIP3 and NIX, it is likely that it influences other mitochondrial protein targets such as TIMM50. Indeed, we have previously demonstrated that TIM50 in yeast is hyperphosphorylated in Δ *pptc7* yeast, and that phosphomimetic mutation of this protein leads to decreased mitochondrial protein import. These observations are currently within the discussion section of our manuscript, and could constitute an alternative mechanism by which PPTC7

influences mitochondrial function. Beyond TIMM50, three other phosphoisoforms reproducibly increase across all model systems tested, and these phosphorylation events may affect their enzymatic activity, protein stability, import into the organelle, protein-protein interactions, or a myriad of other functions. We are indeed interested in understanding the effects of these phosphorylation events and have initiated studies on a handful of these modifications. Alternatively, as the reviewer noted, phosphorylation may dictate mitophagic selectivity of matrix-localized as has been proposed in yeast by the Abeliovich lab. This possibility has been mentioned and relevant citations have been added to the discussion.

Reviewer #4

The work demonstrates the role of the phosphatase Pptc7 in regulating mitochondrial protein content by altering phosphorylation dynamics. The authors also point out the mitophagy receptors Bnip3 and Nix as Pptc7 substrates that undergo direct dephosphorylation. The hypothesis and experiments to test various hypothesis are well laid out. I have a few questions about the phosphoproteomic analysis.

-What was the specificity of the phosphoenrichment? Or what % of the peptide pool were phosphopeptides?

The phosphopeptide-enrichment yielded 13,174 phosphopeptides from 20,571 total peptides—rendering a 64.0% enrichment.

- Line 483: Phosphorylation sites were considered localized if they delivered MaxQuant localization scores >0.75. Is this the cut-off for including or excluding a phosphosite identification? If so, what's the reason for selecting this cut off and do the phosphosites identified (overlapping and unique) get altered if it is set higher, say 0.90?

All phosphorylation site identifications included in the final dataset (n=11,435) have at least a 0.75 localization probability—all sites with lower probabilities were removed. The number of phosphosites meeting a 0.90 localization probability is 9,403. We chose the 0.75 localization probability because this threshold has been defined by the community as “Class I” (Olsen et al., Cell, 2006, PMID: 17081983). A more recent study analyzed a library of synthetic phosphopeptides and reported a site localization error rate of 3.1% within the set of quantified phosphorylation sites possessing at least 0.75 localization probability as determined by MaxQuant (Bekker-Jensen Nat Commun, 2020, PMID: 32034161). Further, our phosphopeptide spectra were collected using double the resolving power that was used to collect the synthetic phosphopeptide spectra from Bekker-Jensen 2020 (60,000 as opposed to 30,000). The 0.75 probability threshold is a well-established standard in the community (Jiang, Mol Cell Proteomics, 2021, PMID: 34737085).

- How were the figures generated? Which software was used for making plots? Including those details would make the Methods section more complete.

We have added details regarding figure generation, statistical analysis, and plot generation in the revised methods section of our manuscript.

REVIEWERS' COMMENTS

Reviewer #1 (Remarks to the Author):

The authors very thoroughly and adequately replied to all the comments raised. The manuscript is now highly improved and I do not have any further comments.

Reviewer #2 (Remarks to the Author):

Comments addressed

Reviewer #3 (Remarks to the Author):

The authors added a substantial amount of information that addressed my comments thoroughly. The paper highlights the importance of managing mitochondrial protein phosphorylation to maintain a healthy mitochondrial population. This is an important study that has substantial physiological relevance and is significant to the field.

Reviewer #4 (Remarks to the Author):

Thank you for addressing my questions regarding phosphoenrichment and adding the details on figure generation. I'm curious if the TiO₂ and FeNTA enriched samples were pooled together before injecting on the mass spectrometer or run separately. It wasn't quite clear from the methods write up. Also, the amount of peptides injected for proteomic analysis is mentioned as 1ug, while volume (4uL) is written for phosphopeptide samples. Could you add the amount injected for phosphopeptides as well? This would make the methods section more clear and complete.